# AI-BAAM: AI-Driven Bank Statement Analytics as Alternative Data for Malaysian MSME Credit Scoring

## Abstract

Despite accounting for 96.1% of all businesses in Malaysia (Department of Statistics Malaysia, 2025), access to financing remains one of the most persistent challenges faced by Micro, Small, and Medium Enterprises (MSMEs). Newly established businesses are often excluded from formal credit markets as traditional underwriting approaches rely heavily on credit bureau data. This study investigates the potential of bank statement data as an alternative data source for credit assessment to promote financial inclusion in emerging markets. First, we propose a cash flow-based underwriting pipeline where we utilize bank statement data for end-to-end data extraction and machine learning credit scoring. Second, we introduce a novel dataset of 611 loan applicants from a Malaysian lending institution. Third, we develop and evaluate credit scoring models based on application information and bank transaction-derived features. Empirical results demonstrate that incorporating bank statement features yields substantial improvements, with our best model achieving an AUROC of 0.806 on in-time validation, representing a 24.6% improvement over models using application information only. Finally, we will release the anonymized bank transaction dataset to facilitate further research on MSME financial inclusion within Malaysia's emerging economy.

## 1 Introduction

Financial inclusion remains a pressing challenge in emerging markets, where a significant portion of the population lacks sufficient credit history to access formal lending. According to the Securities Commission Malaysia (Malaysia, 2025), MSMEs are estimated to contribute around 60% of Malaysia's gross domestic product. Despite their significant contributions, MSMEs remain underserved by financial institutions in terms of access to financing, giving rise to an estimated MYR 90 billion funding gap (Corporation, 2017). One fundamental issue is that many MSMEs do not have lending history. Traditional credit assessment relies heavily on credit bureau data such as repayment history, outstanding obligations, and past delinquencies. While traditional credit models work well for established businesses, they have shortcomings for MSMEs with thin credit files. First, they are inherently backward-looking and focus on past repayment behavior rather than current or forward-looking capacity to repay. Second, they overlook real-time financial signals or operational dynamics that could more accurately reflect a firm's present financial health. Third, they omit alternative indicators of creditworthiness, such as cash flow consistency, receivables and payables patterns, digital transactions, and behavioral metrics from daily financial activity.

On the other hand, bank statements represent an up-to-date and verifiable source of financial behavior that also captures income regularity, spending patterns, and cash flow stability. This study explores the use of Malaysian bank statement transactions as alternative data for credit scoring model development and evaluates their predictive value for MSMEs. The objectives are as follows:

- To propose a cash flow-based underwriting workflow capable of ingesting and analyzing bank transaction data for credit decisioning to narrow the MSME financing gap.

- To introduce the first-ever Malaysian bank statement dataset based on MSME loans.

Figure 1: Proposed end-to-end workflow using bank statement transaction data for credit scoring.

- To evaluate the performance of machine learning–based credit scoring models trained on the proposed bank statement transaction dataset, and to examine the feasibility and predictive power of transaction-derived features in assessing MSME creditworthiness.

## 2  RELATED WORK

With the recent technological advancements in machine learning algorithms, the research community has recognized the limitations of existing traditional credit scoring methods (Gote & Mendhe, 2024). Such studies are crucial for enhancing financial inclusion of MSMEs in emerging markets, where MSMEs are often the backbone of economic growth despite having limited access to financing (Elebe & Imediegwu, 2021). As such, alternative data from non-traditional sources that are not included in standard credit bureau files are being used for credit scoring (Group, 2024).

**Limitations of Traditional Credit Scoring.** Traditional credit scoring models used by financial institutions primarily rely on credit bureau data such as payment history, amount of debt, and other indicators (Shivhare, 2024). Unfortunately, reliance on such data creates a high entry barrier for MSMEs in emerging economies where they usually lack audited financial statements or loan servicing history required by financial institutions (Courchane & Baines, 2020; Elebe & Imediegwu, 2021). This leads MSMEs to be perceived as high-risk applicants and results in a perpetual cycle of financial exclusion that severely limits their growth potential (Courchane & Baines, 2020). Besides, traditional methods fail to capture recent cash flow status of MSMEs, which can be an accurate representation of financial health (Elebe & Imediegwu, 2021). A recent survey indicates that 58% of lending institutions feel less confident to make decisions based on traditional credit data only (Watterson, 2024). The increasing dissatisfaction motivates the urgent need to use alternative data for a more adaptive and inclusive credit assessment (Gote & Mendhe, 2024).

**Credit Scoring with Alternative Data.** In order to overcome the limitations of traditional methods, credit underwriting using alternative data has emerged as a new approach (Gote & Mendhe, 2024; Watterson, 2024; Ngwenya, 2024). For instance, mobile network data are used for credit scoring in Africa (Ngwenya, 2024; Gathu, 2020) and bank account transactions are used for cash flow underwriting in consumer lending (Watterson, 2024). Recent studies suggest that incorporating transactional data can improve predictive accuracy and expand credit to underserved populations (Djeundje et al., 2021). For instance, research has shown that retail transaction data can help construct alternative credit scores (Lee et al., 0). A study examining two Indian financial technology (FinTech) firms found that transactional data from platform activity worked as well as credit bureau data in predicting creditworthiness and improved the predictive accuracy when combined with bureau data (Caire & Vidal, 2024). However, the adoption of alternative data in MSME lending in Malaysia remains at an early stage, with limited research focus from the industry.

**Machine Learning Models in Credit Scoring.** Machine learning (ML) models have proven to be highly effective in processing financial data to produce accurate credit scores (Bücker et al., 2022; Gunnarsson et al., 2021; Trivedi, 2020). These include statistical models like Logistic Regression (LR), Naive Bayes (NB), and ensemble methods like Random Forest, AdaBoost, and Gradient Boosting (Bücker et al., 2022; Trivedi, 2020). For example, a Random Forest model achieved the best performance on Taiwanese credit card data (Abbas & Hussein, 2024). A NB classifier

performed well on a highly imbalanced dataset from a New Zealand lender, especially when undeclared features derived from bank statements were incorporated to supplement application form data (Bunker et al., 2016). Given the positive impact of ML models in credit scoring, we aim to improve financial inclusion of Malaysian MSMEs through proposed cash flow underwriting workflow.

**Agentic Workflow in Credit Scoring.** The deployment of Large Language Model (LLM)-powered agentic systems marks a major advancement in automated credit scoring, loan approval, and customer risk profiling within financial services (Okpala et al., 2025; Ali, 2025; Paleti, 2024). Recent agentic frameworks have demonstrated strong performance in complex model risk management tasks, including credit card approval and portfolio risk modeling (Okpala et al., 2025). Beyond these use cases, agentic workflows are transforming banking by enabling adaptive credit profiling and predictive loan approvals, thereby improving risk assessment accuracy through bias detection and intelligent decision automation (Paleti, 2024). Similar approaches have been explored in mortgage lending, where machine learning-driven agents streamline underwriting and accelerate credit evaluation (Chitturi, 2025). However, existing methods remain limited in production deployment and lack a fully grounded, end-to-end workflow for MSMEs credit scoring.

## 3 BANK STATEMENT CASH FLOW UNDERWRITING

The proposed cash flow underwriting workflow enhances traditional credit underwriting by integrating bank statement-derived features into the credit decision-making process. As illustrated in Figure 1, the workflow consists of six main modules, each serving a distinct purpose. By automating manual extraction and analysis tasks, it shortens turnaround time and improves operational efficiency while expanding credit access for thin-file MSMEs who lack sufficient credit bureau history.

**Key Information Extraction Module.** The key information extraction module employs Optical Character Recognition (OCR) to extract essential fields from bank statements, including the bank name, account number, account holder name, address, and statement date. These attributes are cross-validated across statements to ensure accurate document ownership verification. These fields enable the engine to link multiple accounts belonging to the same business entity, providing a holistic view of its financial activity and cash flow across multiple bank accounts.

**Transaction Table Extraction Module.** The transaction table extraction module localizes and digitizes tabular transaction data using OCR and layout analysis. It identifies table headers, then maps rows and columns to extract transaction date, description, debit, credit, and balance amounts. Additional pre-processing handles merged cells, page breaks, and format inconsistencies to preserve data fidelity. The resulting structured transaction records serve as critical inputs for downstream analytical modules that evaluate transaction patterns and account-level cash flow dynamics.

**Fraud Analysis Module.** The fraud analysis module leverages Computer Vision and rule-based techniques to detect tampered or falsified bank statements. For each document, it examines both visual and metadata cues such as inconsistent font styles or sizes, altered layouts, metadata mismatches, and abnormal pixel-level patterns that may indicate digital editing. Once potential tampering is identified, the affected statements are automatically flagged and returned to bank officers for manual verification. This module provides an additional layer of security and assurance within the underwriting workflow, ensuring that only authentic bank statements are analyzed for credit scoring.

**Network Analysis Module.** The network analysis module constructs transaction networks from the extracted transaction data to uncover relationships among senders and beneficiaries across multiple bank accounts. Using graph-based algorithms, it maps transaction linkages to detect circular fund flows, kiting activities, and transfers to blacklisted or interrelated entities. This module also quantifies network connectivity and transaction density to reveal abnormal money movement patterns that may indicate fraudulent activity or financial distress, thereby enhancing overall risk assessment.

**Cash Flow Analysis Module.** The cash flow analysis module evaluates cash flow status based on the inflows and outflows extracted from transaction records. It then applies Natural Language Processing to infer transaction intent and classify entries into meaningful categories based on their descriptions. Using these classifications, this module groups related transactions to identify spending and receipt patterns, enabling the detection of outliers that deviate from typical cash flow behavior. It also computes key metrics such as average, highest, and lowest balances, quantifying cash flow health and provides predictive features for credit scoring.

**Data & Scoring Layer (Cash Flow Underwriting).** This layer stores the analyzed data and engineered features in a secure database to ensure traceability, regulatory compliance, and auditability. The stored data are then preprocessed, and feature selection methods are applied to retain the most informative variables. Using the selected features, a predictive model is trained to estimate the probability of default and classify credit risk.

# 4 BANK STATEMENT TRANSACTION DATASET

Table 1: Statistics of MSME loan application datasets.

| Split | Non-Event | Event | Total |
|---|---|---|---|
| Train (60% in-time) | 310 | 56 | 366 |
| Validation (40% in-time) | 208 | 37 | 245 |
| Overall | 518 | 93 | 611 |

Transaction data derived from bank statements represents a valuable alternative resource for cash flow underwriting. To the best of our knowledge, no published studies have examined the use of bank statement data for credit assessment of MSMEs in Malaysia. In collaboration with a lending institution, we constructed the first Malaysian bank statements dataset to address this gap.

This study adopts the Cross-Industry Standard Process for Data Mining (CRISP–DM) framework (Chapman et al., 2000) as the methodological basis. CRISP–DM is a widely recognized process model for systematic FinTech studies (Cheng, 2023; Rawat, 2023; da Rocha & de Sousa Junior, 2010) and comprises six main phases, i.e., business understanding, data understanding, data preparation, modeling, evaluation, and deployment. The first phase involves evaluating bank statement transaction data as an alternative source for MSME credit risk assessment in Malaysia. The second phase focuses on constructing the proposed dataset of 611 MSME loan applicants, which is partitioned using a temporal validation strategy as shown in Table 1.

**Dataset Distribution and Preparation.** The dataset is split into training (60%) and validation (40%) sets using stratified random sampling to preserve class distribution. Among all applicants, 518 have a good credit record, while 93 have a history of default. Each applicant's record contains two main components: application form information submitted during the loan application process (e.g., demographic and business characteristics) and bank statement transaction data capturing detailed inflows and outflows over a six-month period. The third phase involves data preparation and includes several key tasks: (1) cleaning and de-duplicating data to ensure consistency, (2) fixing missing values, standardizing transaction categories, and validating data integrity across all records, and (3) deriving features and variables from transaction data, such as determining cash flow stability, deposit regularity, and balance volatility.

**Dataset Masking.** All personally identifiable information was anonymized prior to analysis. To preserve confidentiality, the features are grouped into application information (7 features) and bank statement features (10 features). The specific feature calculations cannot be disclosed due to a non-disclosure agreement and all bank statement features are derived solely from bank statement data.

# 5 TRANSACTION DATA-BASED CREDIT SCORING

The fourth phase of CRISP–DM in this study employs Logistic Regression on application form and bank transaction data derived from our dataset as the baseline credit scoring model. LR is widely adopted in credit risk modeling due to its interpretability, statistical robustness, and capacity to estimate the probability of default (Bücker et al., 2022; Trivedi, 2020; Abbas & Hussein, 2024; Bunker et al., 2016). To ensure that only informative predictors are retained, feature selection and transformation are guided by the Weight of Evidence (WOE) and Information Value (IV) framework, which are widely used in credit risk modeling for quantifying the predictive power of individual variables (Ngwenya, 2024). In practice, each feature is divided into discrete intervals based on suitable thresholds determined from the data. These bins allow the calculation of WOE and IV at the group level, which makes them easier to interpret within a LR model. The following notation is used to quantify the distribution of defaults and non-defaults across feature bins:

Let $y_i \in \{0, 1\}$ indicate default ($y_i = 1$) or non-default ($y_i = 0$) for applicant $i$. The total number of default and non-default applicants is defined as:

$$N_b = \sum_{i=1}^{n} \mathbb{I}(y_i = 1), \qquad\qquad N_g = \sum_{i=1}^{n} \mathbb{I}(y_i = 0), \qquad (1)$$

Furthermore, let $x_{ij}$ denote the value of feature $j$ for applicant $i$. For a given feature $j$, suppose it is divided into $K_j$ disjoint bins $\{B_{j1}, \ldots, B_{jK_j}\}$; then the corresponding counts within each bin are defined as:

$$n_{bjk} = \sum_{i=1}^{n} \mathbb{I}(x_{ij} \in B_{jk}) \, \mathbb{I}(y_i = 1),$$

$$n_{gjk} = \sum_{i=1}^{n} \mathbb{I}(x_{ij} \in B_{jk}) \, \mathbb{I}(y_i = 0). \qquad (2)$$

The distribution of defaults and non-defaults in each bin is:

$$\text{Dist}_{jk}^{(b)} = \frac{n_{bjk}}{N_b}, \qquad\qquad \text{Dist}_{jk}^{(g)} = \frac{n_{gjk}}{N_g}. \qquad (3)$$

**Weight of Evidence (WOE).** Firstly, we calculate the WOE for bin $k$ of feature $j$ using:

$$\text{WOE}_{jk} = \log\left(\frac{\text{Dist}_{jk}^{(g)}}{\text{Dist}_{jk}^{(b)}}\right) = \log\left(\frac{n_{gjk}/N_g}{n_{bjk}/N_b}\right). \qquad (4)$$

A positive $\text{WOE}_{jk}$ indicates that bin $B_{jk}$ is more common among non-default cases, suggesting lower risk, whereas a negative value indicates a higher likelihood of default. Once WOE is computed for each bin, the overall predictive strength of feature $j$ can then be summarized by its IV.

**Information Value (IV).** Secondly, we measure the IV of feature $j$ by aggregating the WOE across all respective bins:

$$\text{IV}_j = \sum_{k=1}^{K_j} \left(\text{Dist}_{jk}^{(g)} - \text{Dist}_{jk}^{(b)}\right) \text{WOE}_{jk}. \qquad (5)$$

A larger $\text{IV}_j$ indicates stronger discriminatory power of feature $j$ between default and non-default classes. In this study, IV serves as both a feature-ranking criterion and an interpretability measure, helping to identify the most influential transaction-level indicators for creditworthiness. Following industry practice (Siddiqi, 2017), we adopt the commonly used interpretive thresholds: IV $< 0.02$ (not predictive), $0.02 \leq \text{IV} < 0.1$ (weak), $0.1 \leq \text{IV} < 0.3$ (medium), $0.3 \leq \text{IV} < 0.5$ (strong), and IV $\geq 0.5$ (suspiciously high; suggesting potential data leakage).

**Binning of Bank Statement Features.** Specifically, we use quantile or supervised monotonic binning to obtain $\{B_{jk}\}_{k=1}^{K_j}$ for continuous transaction-derived variables (e.g., cash flow stability, balance volatility). Then, we group rare datapoints to ensure each bin has sufficient defaults and non-defaults for sparse categorical variables (e.g., State/Location). Computations of WOE and IV are performed on training folds only to avoid data leakage.

## 6  EXPERIMENTS

Following the fifth phase of CRISP–DM, we conduct a series of experiments to evaluate the effectiveness of the proposed cash flow underwriting workflow and the role of bank statement data in enhancing MSME credit scoring performance. The experiments include comparative analyses with established machine learning methods and detailed ablation studies on transaction-derived features.

### 6.1  IMPLEMENTATION DETAILS

We benchmark the baseline LR model against several widely used machine learning methods, including Random Forest (RF) (Breiman, 2001), Gradient Boosting (GB) (Friedman, 2001), and

AdaBoost (AB) (Schapire, 2013). All models are implemented in scikit-learn (scikit learn, 2025c;a;b;d). For the results in Section 6.2, models were trained with default hyperparameters. In Section 6.3, hyperparameters were tuned using a randomized grid search with 50 trials to ensure robustness. We evaluate model performance using the Area Under the Receiver Operating Characteristic Curve (AUROC) (Bradley, 1997), which measures the ability to discriminate between default and non-default cases across varying thresholds. An AUROC of 0.5 indicates no discriminative power (equivalent to random guessing), whereas a value of 1.0 indicates perfect discrimination. To ensure interpretability and systematic analysis, both application information and bank statement features are grouped into two categories:

i Account Behavior: logarithmic growth rate of average balance, six-month average account balance, change in 3-month minimum balance (recent), percent change in minimum balance vs prior period, recent 3-month maximum average balance, and credit repayment capacity.

ii Business Demographics: business operational duration, geographic region, industry classification (MSIC), total board directors, and minimum director age.

This structured feature grouping facilitates clearer attribution of model performance to different behavioral and demographic dimensions of MSME credit profiles. It also supports experimental reproducibility, while respecting confidentiality constraints around proprietary feature derivations.

## 6.2 Quantitative Results

**In-Time Validation Performance.** Quantitative results on the in-time validation split are presented in Figure 2. The blended feature approach with Logistic Regression achieves the highest validation AUROC of 0.806, demonstrating the complementary nature of application information and bank statement features. This represents a 18.5% improvement over AdaBoost with blended features (0.680 validation AUROC). Bank statement features alone yield strong performance (LR: 0.763), outperforming application info alone (LR: 0.647) across all models. Notably, Logistic Regression consistently achieves superior or competitive validation AUROC compared to ensemble methods (GB: 0.599, RF: 0.651, AB: 0.645), which is particularly pronounced when operating on smaller validation sets under class imbalance (310 non-event vs. 56 event cases in the training set). In such settings, models providing well-calibrated probabilities and strong generalization tend to perform better, a finding consistent with prior benchmarking studies showing that LR often matches or outperforms tree-based ensembles on credit datasets (Brown & Mues, 2012; Lessmann et al., 2015).

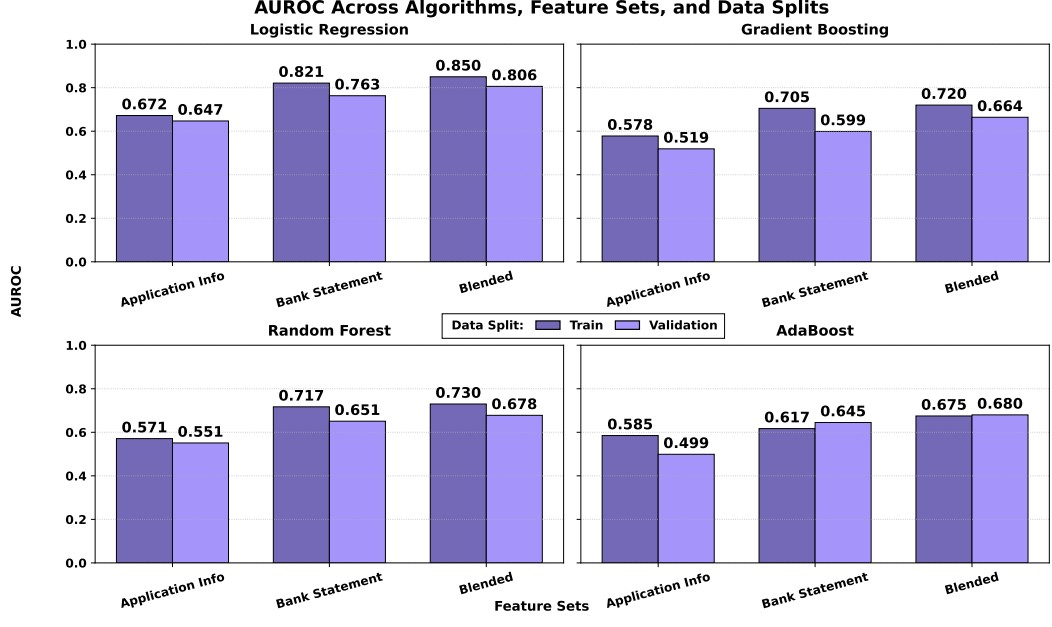

Figure 2: Evaluation results of all models across different feature combinations and data splits.

Table 2: Summary of extraction performance, latency, and cost for methods evaluated on key information and transaction table extraction tasks. Accuracy scores are averaged over five fields (key info) and five column types (table); F1 is averaged over six banks. Latency is the average total time per document (seconds) and for the proposed template matching method was measured on a Mac-Book Air 2025 with M4 chip and 24GB unified memory. Cost is the total API token cost per batch. Best results are in **bold**; second-best are underlined. Full breakdowns in Appendix Tables 3–14.

| Method | Key Information | | Table Extraction | | | Latency (s) | | Cost (USD) |
| | Exact Match | NED Score | Match. NED | Exact NED | F1 Score | Key Info | Table Extraction | Table Extraction |
| --- | --- | --- | --- | --- | --- | --- | --- | --- |
| docTR + GPT-4o-mini | 91.87 | 99.20 | 88.41 | 78.52 | 92.16 | 2.38 | 7.59 | — |
| docTR + GPT-4.1 | 94.79 | 99.32 | 94.90 | 94.45 | 97.51 | 6.13 | 11.92 | 0.53 |
| PyMuPDF + GPT-4o-mini | 89.32 | 92.22 | 90.34 | 74.78 | 90.91 | 1.86 | 4.98 | 0.04 |
| PyMuPDF + GPT-4.1 | 93.23 | 93.86 | 94.76 | 93.49 | 97.64 | 0.81 | 10.72 | — |
| Pdfium + GPT-4o-mini | 87.65 | 92.01 | 89.33 | 78.20 | 89.63 | 1.86 | 6.21 | — |
| Pdfium + GPT-4.1 | 93.57 | 94.20 | 93.82 | 89.63 | 95.66 | 0.86 | 11.19 | — |
| **Ours (template matching)** | **100.00** | **100.00** | **98.08** | **97.80** | **100.00** | **0.01** | **0.11** | $0 |

**Extraction Module Performance.** Beyond credit scoring, we evaluate the upstream AI modules responsible for bank statement data extraction, as their accuracy directly affects downstream feature quality. For key information extraction, we benchmark 15 OCR-GPT pipeline configurations against our template matching approach across six Malaysian banks (Table 3 in Appendix). Our method achieves perfect 100% exact match accuracy and Normalized Edit Distance (NED) score across all five extraction fields (bank name, account holder, account number, address, and statement date), whereas the best-performing baseline (Azure AI 4.0 + GPT-4.1) reaches 100% on four fields but drops to 94.82% on address extraction. For transaction table extraction, we evaluate 16 configurations including recent vision-language models (Tables 8 and 9). Our template matching approach achieves the highest average matching NED of 98.08% and exact NED of 97.80% across all column types. The closest competitor, Surya Paruchuri & Team (2025) + docTR Mindee (2021) with GPT-5-mini, attains 96.94% and 94.23% respectively. Notably, end-to-end vision models such as olmOCR (Poznanski et al., 2025) and paddleOCR-VL (Cui et al., 2025a) show substantially lower performance on structured table extraction, underscoring the advantage of specialized pipeline designs for financial documents. These general-purpose models struggle with the highly structured yet bank-specific formatting of Malaysian bank statements, particularly multi-line transaction descriptions, inconsistent date formats, and merged table cells that vary across institutions. Our template matching method succeeds because it exploits the deterministic and standardized layouts within each bank's PDF format, bypassing the need for LLM entirely and thus eliminating both error propagation and API dependency. Table 2 summarizes the performance of selected configurations across both extraction tasks. Full per-bank breakdowns, F1 scores, and detailed analysis along with a comprehensive latency and cost analysis across all configurations are provided in Appendix A.5.

**Cost and Latency Analysis.** Operational efficiency is a key consideration for AI-BAAM. We report latency and cost measurements across all configurations in Appendix Tables 12, 13, and 14. For key information extraction, our template matching method averages 0.01s per document with perfect accuracy, orders of magnitude faster than the fastest LLM-based baseline (PyMuPDF (pymupdf, 2025) + GPT-4.1 at 0.81s). For table extraction, our method averages 0.11s compared to 4.98s for PyMuPDF + GPT-4o-mini. For transaction table extraction, text-based pipelines such as PyMuPDF + GPT-4o-mini average under 5s, whereas end-to-end vision-language models require 104–150s. Among LLM-based configurations, Surya + docTR with GPT-5-mini ($0.02) and docTR with GPT-4.1-nano ($0.03) are the most cost-effective, while docTR + GPT-4.1 incurs $0.53 per batch. Our template matching approach eliminates API costs entirely while maintaining sub-second processing latency, achieving the best accuracy–efficiency trade-off for production deployment.

## 6.3 ABLATION STUDIES

Results in Figure 3 presents the information value of individual features, ranked by discriminatory power. Bank statement features dominate the top positions with *logarithmic growth rate of average*

*balance* having the highest score of 0.484. In contrast, the strongest application information feature is *business operational duration* with an IV of 0.213, ranking far below the transaction-based features. These findings reinforce the superior discriminatory strength of bank statement features in distinguishing default from non-default outcomes.

Furthermore, we conducted ablation experiments by progressively removing feature groups to study which features matter most for MSMEs credit scoring. We applied 5-fold cross-validation on the in-time training split (366 samples) of the proposed dataset to reduce the risk that results are due to random chance and to increase confidence in model stability. From results illustrated in Figure 2, we observe that Logistic Regression with blended features achieves the strongest validation AUROC of 0.850, outperforming Gradient Boosting (0.720) and Random Forest (0.730). When considering bank statement features alone, Logistic Regression achieves a validation AUROC of 0.821, compared to 0.672 for account information alone, demonstrating a substantial uplift of 0.149 points (approximately 22% relative improvement). These results confirm that transactional data provides significant incremental predictive value. Across all feature configurations and data splits, Logistic Regression consistently outperforms tree-based ensembles. Moreover, models incorporating bank transaction data consistently outperform those relying on application information alone. Further studies of the credit score distribution for each feature groups as well as the analysis on rejected cases can be found in Appendix A.6.1-A.6.2.

In general, these findings strongly support the hypothesis that bank statement transaction data offers significant predictive power in credit risk assessment for MSMEs in Malaysia. While application form information provides some discriminatory power, the inclusion of bank transaction-based features substantially enhances model performance, reinforcing the effectiveness of AI-BAAM.

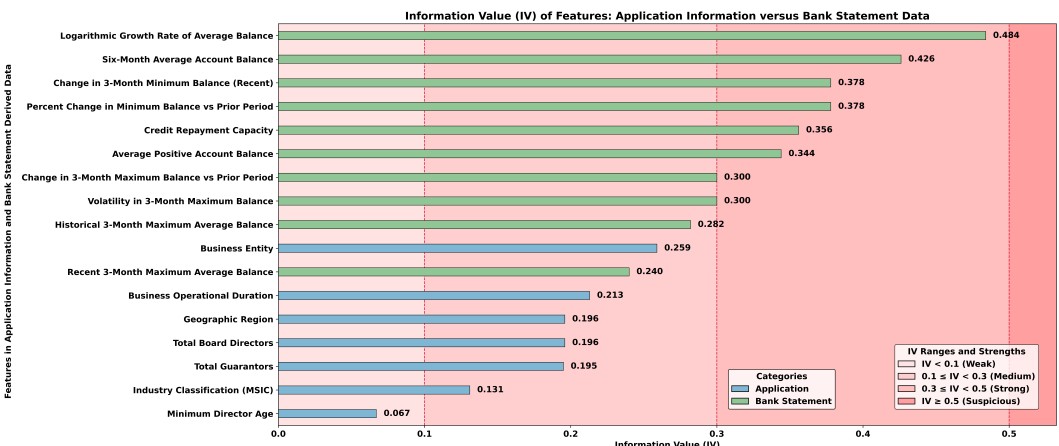

Figure 3: IV of features derived from application information and bank statement data.

## 7 CONCLUSION

This study introduces bank statement transactions as alternative data for MSME credit scoring in Malaysia and presents AI-BAAM, an end-to-end cash flow underwriting workflow spanning document extraction, feature engineering, and predictive modeling. Empirical results confirm that transaction-derived features capture dynamic financial behavior overlooked by traditional credit models: models trained on bank transaction data alone substantially outperform those using only application information, and combining both feature sets yields the highest predictive power. Beyond credit scoring, we benchmark over 30 OCR and LLM configurations for key information and transaction table extraction across six Malaysian banks. Our template matching approach achieves perfect accuracy on key information fields and the highest NED and F1 scores on table extraction, while processing documents in under 0.12 seconds with zero API cost. In contrast, the best LLM-based pipelines require 0.81–11.92 seconds per document and incur up to $0.53 per batch, and end-to-end vision-language models exhibit 104–150 seconds latency with substantially lower extraction quality. These findings demonstrate that our template matching method offers a superior accuracy-efficiency trade-off over general-purpose LLM pipelines for Malaysian bank statements.

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

# A   APPENDIX

## A.1   LOGISTIC REGRESSION MODEL

Given that WOE provides a log-odds transformation of each feature, this aligns naturally with the LR model. The WOE transformed feature value can be represented as $\text{WOE}_j(x_{ij})$ and the LR model can be expressed in a simplified form as:

$$\log \frac{P(y_i = 1 \mid \mathbf{x}_i)}{P(y_i = 0 \mid \mathbf{x}_i)} = \beta_0 + \sum_{j=1}^{d} \beta_j \, \text{WOE}_j(x_{ij}). \tag{6}$$

where $\beta_0$ is the intercept and $\beta_j$ represents the coefficient for feature $j$. This allows direct interpretation of coefficients in terms of the credit risk associated with each feature. Positive $\beta_j$ values indicate that higher WOE corresponds to lower default risk, and vice versa. Hence, WOE encoding stabilizes estimation and supports consistent, monotonic relationships between predictors and credit outcomes.

Formally, let $\mathbf{x}_i \in \mathbb{R}^d$ denote the feature vector for applicant $i$, where $d$ is the number of features engineered from both application form data and bank statement transactions, and let $y_i \in \{0, 1\}$ be the binary output indicating default ($y_i = 1$) or non-default ($y_i = 0$). The LR model specifies the conditional probability of default as

$$P(y_i = 1 \mid \mathbf{x}_i; \boldsymbol{\beta}) = \sigma(\beta_0 + \mathbf{x}_i^\top \boldsymbol{\beta}), \tag{7}$$

where $\sigma(z) = \frac{1}{1+e^{-z}}$ is the sigmoid function, $\beta_0 \in \mathbb{R}$ is the intercept term, and $\boldsymbol{\beta} \in \mathbb{R}^d$ is the coefficient vector associated with the predictors. The parameters are estimated by maximizing the penalized log-likelihood function:

$$\mathcal{L}(\boldsymbol{\beta}) = \sum_{i=1}^{n} [y_i \log p_i + (1 - y_i) \log(1 - p_i)] - \lambda \, \|\boldsymbol{\beta}\|_2^2, \tag{8}$$

where $p_i = P(y_i = 1 \mid \mathbf{x}_i; \boldsymbol{\beta})$ and $\lambda \geq 0$ controls the strength of the $\ell_2$ regularization (Ng, 2004). This mitigates overfitting by shrinking coefficient magnitudes, which is particularly important when modeling high-dimensional, transaction-derived features.

## A.2   USE OF LARGE LANGUAGE MODELS

### A.2.1   DATASET

We gathered bank statements from various financial institutions in Malaysia. The dataset consists of approximately 110 unique bank statement PDFs from the top 6 banks in Malaysia: Maybank, Public Bank, CIMB Bank, RHB Bank, Hong Leong Bank, and AmBank. Each bank statement contains key information fields such as bank name, account holder name, account number, address, and statement date, as well as transaction tables listing individual transactions with details such as date, description, debit amount, credit amount, and balance.

### A.2.2   MODEL CHOICES

Precise text bounding box coordinates for OCR results are required because bank officers must cross-check extraction results before they can be used for subsequent processing; this ensures transparency, traceability, and auditability of our AI results, and explains our design choice of using OCR and LLMs for financial information extraction. The OCR models used are mainly from Azure with different variations: prebuilt-bankStatement.us is an Azure Document Intelligence prebuilt OCR model specifically designed for bank statements in the US, while pretrained-read is a more general OCR model that can be fine-tuned for various document types. The Azure Vision 4.0 model refers to the Azure generic OCR model. The GPT models used are GPT-4o-mini and GPT-4.1, which are different versions of OpenAI's GPT-4 architecture with varying capabilities and performance levels.

Certain specialized OCR models such as MinerU (Wang et al.), PyMuPDF (pymupdf, 2025) and PPStructureV3 (Cui et al., 2025b) are included for comparison since they are widely adopted in the industry for document processing tasks. PyMuPDF-Formatted refers to formatting the output from document parser to imitate actual document layout before passing them into LLM. LLM-based OCR models such as olmOCR (Poznanski et al., 2025), paddleOCR-VL (Cui et al., 2025a), and Surya (Paruchuri & Team, 2025) are also included for table extraction comparisons. Our template matching method is built based on the layout of the bank statements in our dataset, where we manually define the regions of interest for key information fields and transaction tables. For our method, we use DB with ResNet50 backbone (Liao et al., 2020) for text detection and CRNN with VGG16 backbone (Shi et al., 2016) for text recognition. Then we build our own template matching algorithm in Python logic. Do note that we are the only method that is not using any LLMs for information extraction.

As shown in Tables 3, 4, 8, and 9, our template matching method achieves the best performance when the bank statement formats are consistent, which is the case in Malaysia. In real-world scenarios where bank statement formats can vary significantly, template matching methods may not be as effective as more flexible OCR models. In future work, we plan to explore more advanced techniques such as OCR+LLM-based layout analysis and document understanding models (Deepseek-OCR 2 (Wei et al., 2026), MonkeyOCR (Li et al., 2025)) to improve the robustness of key information and table extraction across diverse bank statement formats.

### A.2.3 EVALUATION METRIC

Key information results are evaluated based on exact match accuracy between the extracted text and the ground truth for each field. An exact match is counted when the extracted text matches the ground truth character by character, including spaces and punctuation. The accuracy is then calculated as the percentage of exact matches over the total number of key information fields evaluated.

Table extraction results are evaluated based on the normalized edit distance (NED) between the extracted text and the ground truth for each transaction row. The NED for a single row entry is calculated as follows:

$$\text{NED} = 1 - \frac{\text{Edit Distance}}{\max(\text{Length of Extracted Text}, \text{Length of Ground Truth Text})} \quad (9)$$

where Edit Distance is the minimum number of operations (insertions, deletions, substitutions) required to transform the extracted text into the ground truth text. A higher NED value indicates better accuracy, with a maximum of 1.0 representing a perfect match. The score is then scaled to the range of 0 to 100 for easier interpretation.

We report two variants of the per-column similarity score to capture different aspects of extraction quality:

**Matching NED Score (%).** For each column (date, description, debit, credit, balance), we compute the NED for every matched row. Only rows with NED $> 70\%$ are retained, and the per-column score is the average NED across these filtered rows for a given bank. This metric measures how accurately the system extracts content for rows that it has reasonably identified, by excluding severely misaligned or garbled entries that fall below the 70% threshold. It thus reflects extraction quality conditioned on successful row matching.

**Exact NED Score (%).** For each column, we compute the NED for every row without any filtering threshold. All rows are included regardless of their NED value, and the per-column score is the average NED across all rows for a given bank. This metric provides a stricter evaluation that penalizes both character-level extraction errors and row-level alignment failures (e.g., missed rows, extra rows, or misaligned entries), as these contribute low NED values that reduce the overall average.

The key difference between the two metrics lies in their treatment of poorly matched rows. Matching NED isolates extraction accuracy by filtering out rows with NED $\leq 70\%$, while exact NED captures the full picture including row detection failures. A large gap between matching and exact NED scores for a given configuration indicates that the system produces accurate extractions when it correctly identifies rows, but frequently fails at row-level alignment or misses rows entirely. Conversely, similar matching and exact NED scores suggest consistent row detection with few alignment failures.

## A.3 QUANTITATIVE RESULTS FOR KEY INFORMATION EXTRACTION

We evaluate key information extraction across the top six largest Malaysian banks: Maybank (MBBE), Public Bank (PBBE), CIMB Bank (CIBB), RHB Bank (RHBB), Hong Leong Bank (HLBB), and AmBank (ARBK). The evaluation covers five structured fields: bank name, account holder, account number, address, and statement date. We report both exact match accuracy with strict character-level equality and Normalized Edit Distance (NED) similarity with a tolerance for minor character-level discrepancies. Tables 3 and 4 summarize the average exact match accuracy and NED score performance across all banks respectively, while Tables 5 and 6 provide per-bank breakdowns.

**Overall Performance.** Our template matching approach achieves a perfect 100% on both exact match accuracy and NED score across all five fields and all six banks, demonstrating robust and consistent extraction without reliance on external OCR or LLM APIs. Among the baselines, configurations pairing high-quality OCR engines (Pretrained-Read, Azure AI 4.0) with GPT-4.1 achieve near-perfect results, with Azure AI 4.0 + GPT-4.1 reaching 100% exact match accuracy on four fields (bank name, account holder, account number, and statement date) and 94.82% on address. The address field proves most challenging across all configurations due to multi-line formatting and bank-specific layout variations.

**OCR Engine Impact.** The choice of OCR engine substantially influences downstream extraction quality. Cloud-based OCR solutions (Pretrained-Read, Azure AI 4.0) consistently outperform open-source alternatives. MinerU exhibits the weakest bank name extraction (28.33%–41.30% exact match accuracy) due to its inability to parse graphical logos and stylized text commonly found in Malaysian bank statement headers. Text-based extractors (PyMuPDF, Pdfium) show variable bank name accuracy (58.76%–74.87%), as they rely on the PDF text layer, which may not encode bank names embedded in header images. In contrast, docTR achieves 100% bank name accuracy by leveraging deep learning-based text detection on rendered page images.

**GPT Model Impact.** Across all OCR engines, upgrading from GPT-4o-mini to GPT-4.1 consistently improves extraction accuracy, particularly for address and statement date fields. For example, with docTR, the address exact accuracy improves from 73.97% to 80.82% when switching to GPT-4.1. This improvement stems from GPT-4.1's enhanced ability to parse multi-line address formats and normalize date representations across different bank templates.

**Per-Bank Variation.** AmBank and Hong Leong Bank statements are generally easier to process due to their standardized digital-native PDF formats, achieving near-perfect scores across most configurations. RHB Bank and CIMB Bank present greater challenges: RHB Bank's multi-column layouts reduce MinerU's performance to 40–70% across most fields, while CIMB Bank's header formatting leads to lower bank name recognition for text-based extractors. These per-bank variations underscore the importance of bank-specific evaluation when deploying extraction systems in multi-bank production environments.

Table 3: Average exact match accuracy scores across all top six largest banks.

| OCR | GPT | Key Information Extraction - Exact Match Accuracy | | | | |
|---|---|---|---|---|---|---|
| | | Bank Name | Acc Holder | Acc Number | Address | Statement Date |
| Prebuilt-BankStatement | N/A | 93.39 | 80.82 | 92.86 | 92.77 | 77.51 |
| Pretrained-Read | gpt-4o-mini | 100.00 | 96.48 | 100.00 | 89.10 | 97.62 |
| Pretrained-Read | gpt-4.1 | 100.00 | 100.00 | 100.00 | 92.43 | 100.00 |
| Azure AI 4.0 | gpt-4o-mini | 100.00 | 98.15 | 100.00 | 88.20 | 97.62 |
| Azure AI 4.0 | gpt-4.1 | 100.00 | 100.00 | 100.00 | 94.82 | 100.00 |
| docTR | gpt-4o-mini | 100.00 | 87.78 | 100.00 | 73.97 | 97.62 |
| docTR | gpt-4.1 | 100.00 | 93.15 | 100.00 | 80.82 | 100.00 |
| MinerU | gpt-4o-mini | 28.33 | 86.48 | 90.95 | 61.43 | 95.00 |
| MinerU | gpt-4.1 | 41.30 | 86.67 | 93.33 | 72.14 | 95.00 |
| PyMuPDF | gpt-4o-mini | 62.09 | 96.30 | 100.00 | 90.58 | 97.62 |
| PyMuPDF | gpt-4.1 | 71.16 | 98.33 | 100.00 | 96.67 | 100.00 |

*Continued on next page*

Table 3 – *Continued from previous page*

| OCR | GPT | Key Information Extraction - Exact Match Accuracy | | | | |
|-----|-----|-----------|------------|------------|---------|----------------|
|     |     | **Bank Name** | **Acc Holder** | **Acc Number** | **Address** | **Statement Date** |
| PyMuPDF-Formatted | gpt-4o-mini | 61.92 | 98.15 | 100.00 | 92.25 | 100.00 |
| PyMuPDF-Formatted | gpt-4.1 | 74.87 | 98.15 | 100.00 | 96.48 | 100.00 |
| Pdfium | gpt-4o-mini | 58.76 | 96.30 | 100.00 | 83.17 | 100.00 |
| Pdfium | gpt-4.1 | 71.16 | 100.00 | 100.00 | 96.67 | 100.00 |
| **Ours (template matching)** | | 100.00 | 100.00 | 100.00 | 100.00 | 100.00 |

Table 4: Average NED scores across all top six largest banks.

| OCR | GPT | Key Information Extraction - NED Score | | | | |
|-----|-----|-----------|------------|------------|---------|----------------|
|     |     | **Bank Name** | **Acc Holder** | **Acc Number** | **Address** | **Statement Date** |
| Prebuilt-BankStatement | N/A | 93.39 | 82.75 | 95.11 | 98.40 | 79.13 |
| Pretrained-Read | gpt-4o-mini | 100.00 | 97.36 | 100.00 | 96.30 | 99.70 |
| Pretrained-Read | gpt-4.1 | 100.00 | 100.00 | 100.00 | 99.34 | 100.00 |
| Azure AI 4.0 | gpt-4o-mini | 100.00 | 98.59 | 100.00 | 96.90 | 99.70 |
| Azure AI 4.0 | gpt-4.1 | 100.00 | 100.00 | 100.00 | 99.59 | 100.00 |
| docTR | gpt-4o-mini | 100.00 | 98.44 | 100.00 | 97.84 | 99.70 |
| docTR | gpt-4.1 | 100.00 | 98.02 | 100.00 | 98.60 | 100.00 |
| MinerU | gpt-4o-mini | 37.00 | 88.29 | 92.74 | 77.46 | 97.50 |
| MinerU | gpt-4.1 | 43.79 | 86.67 | 93.33 | 84.20 | 96.25 |
| PyMuPDF | gpt-4o-mini | 64.99 | 97.28 | 100.00 | 99.11 | 99.70 |
| PyMuPDF | gpt-4.1 | 71.17 | 98.33 | 100.00 | 99.82 | 100.00 |
| PyMuPDF-Formatted | gpt-4o-mini | 69.36 | 98.15 | 100.00 | 99.17 | 100.00 |
| PyMuPDF-Formatted | gpt-4.1 | 75.77 | 99.32 | 100.00 | 99.50 | 100.00 |
| Pdfium | gpt-4o-mini | 63.96 | 97.13 | 100.00 | 98.97 | 100.00 |
| Pdfium | gpt-4.1 | 71.17 | 100.00 | 100.00 | 99.82 | 100.00 |
| **Ours (template matching)** | | 100.00 | 100.00 | 100.00 | 100.00 | 100.00 |

Table 5: Key information extraction exact match accuracy score grouped by bank across OCR and GPT configurations.

| OCR | GPT | Key Information Extraction - Exact Match Accuracy | | | | |
|-----|-----|-----------|------------|------------|---------|----------------|
|     |     | **Bank Name** | **Acc Holder** | **Acc Number** | **Address** | **Statement Date** |
| **CIMB Bank (7 Unique PDFs)** | | | | | | |
| Prebuilt-BankStatement | N/A | 71.43 | 71.43 | 57.14 | 85.71 | 42.86 |
| Pretrained-Read | gpt-4o-mini | 100.00 | 100.00 | 100.00 | 85.71 | 85.71 |
| Pretrained-Read | gpt-4.1 | 100.00 | 100.00 | 100.00 | 85.71 | 100.00 |
| Azure AI 4.0 | gpt-4o-mini | 100.00 | 100.00 | 100.00 | 71.43 | 85.71 |
| Azure AI 4.0 | gpt-4.1 | 100.00 | 100.00 | 100.00 | 100.00 | 100.00 |
| docTR | gpt-4o-mini | 100.00 | 100.00 | 100.00 | 57.14 | 85.71 |
| docTR | gpt-4.1 | 100.00 | 100.00 | 100.00 | 57.14 | 100.00 |
| MinerU | gpt-4o-mini | 0.00 | 100.00 | 85.71 | 85.71 | 100.00 |
| MinerU | gpt-4.1 | 0.00 | 100.00 | 100.00 | 100.00 | 100.00 |
| PyMuPDF | gpt-4o-mini | 71.43 | 100.00 | 100.00 | 85.71 | 85.71 |
| PyMuPDF | gpt-4.1 | 71.43 | 100.00 | 100.00 | 100.00 | 100.00 |
| PyMuPDF-Formatted | gpt-4o-mini | 71.43 | 100.00 | 100.00 | 85.71 | 100.00 |

Table 5 – *Continued from previous page*

| OCR | GPT | Key Information Extraction - Exact Match Accuracy | | | | |
| --- | --- | --- | --- | --- | --- | --- |
| | | Bank Name | Acc Holder | Acc Number | Address | Statement Date |
| PyMuPDF-Formatted | gpt-4.1 | 71.43 | 100.00 | 100.00 | 100.00 | 100.00 |
| Pdfium | gpt-4o-mini | 71.43 | 100.00 | 100.00 | 85.71 | 100.00 |
| Pdfium | gpt-4.1 | 71.43 | 100.00 | 100.00 | 100.00 | 100.00 |
| **Ours (template matching)** | | 100.00 | 100.00 | 100.00 | 100.00 | 100.00 |
| **RHB Bank (10 Unique PDFs)** | | | | | | |
| Prebuilt-BankStatement | N/A | 100.00 | 90.00 | 100.00 | 70.00 | 100.00 |
| Pretrained-Read | gpt-4o-mini | 100.00 | 100.00 | 100.00 | 90.00 | 100.00 |
| Pretrained-Read | gpt-4.1 | 100.00 | 100.00 | 100.00 | 80.00 | 100.00 |
| Azure AI 4.0 | gpt-4o-mini | 100.00 | 100.00 | 100.00 | 100.00 | 100.00 |
| Azure AI 4.0 | gpt-4.1 | 100.00 | 100.00 | 100.00 | 80.00 | 100.00 |
| docTR | gpt-4o-mini | 100.00 | 70.00 | 100.00 | 80.00 | 100.00 |
| docTR | gpt-4.1 | 100.00 | 70.00 | 100.00 | 70.00 | 100.00 |
| MinerU | gpt-4o-mini | 70.00 | 50.00 | 70.00 | 40.00 | 70.00 |
| MinerU | gpt-4.1 | 70.00 | 50.00 | 70.00 | 40.00 | 70.00 |
| PyMuPDF | gpt-4o-mini | 90.00 | 100.00 | 100.00 | 80.00 | 100.00 |
| PyMuPDF | gpt-4.1 | 100.00 | 100.00 | 100.00 | 80.00 | 100.00 |
| PyMuPDF-Formatted | gpt-4o-mini | 100.00 | 100.00 | 100.00 | 100.00 | 100.00 |
| PyMuPDF-Formatted | gpt-4.1 | 100.00 | 100.00 | 100.00 | 100.00 | 100.00 |
| Pdfium | gpt-4o-mini | 70.00 | 100.00 | 100.00 | 80.00 | 100.00 |
| Pdfium | gpt-4.1 | 100.00 | 100.00 | 100.00 | 80.00 | 100.00 |
| **Ours (template matching)** | | 100.00 | 100.00 | 100.00 | 100.00 | 100.00 |
| **Hong Leong Bank (9 Unique PDFs)** | | | | | | |
| Prebuilt-BankStatement | N/A | 100.00 | 88.89 | 100.00 | 100.00 | 100.00 |
| Pretrained-Read | gpt-4o-mini | 100.00 | 88.89 | 100.00 | 100.00 | 100.00 |
| Pretrained-Read | gpt-4.1 | 100.00 | 100.00 | 100.00 | 100.00 | 100.00 |
| Azure AI 4.0 | gpt-4o-mini | 100.00 | 88.89 | 100.00 | 88.89 | 100.00 |
| Azure AI 4.0 | gpt-4.1 | 100.00 | 100.00 | 100.00 | 100.00 | 100.00 |
| docTR | gpt-4o-mini | 100.00 | 66.67 | 100.00 | 88.89 | 100.00 |
| docTR | gpt-4.1 | 100.00 | 88.89 | 100.00 | 100.00 | 100.00 |
| MinerU | gpt-4o-mini | 0.00 | 88.89 | 100.00 | 100.00 | 100.00 |
| MinerU | gpt-4.1 | 44.44 | 100.00 | 100.00 | 100.00 | 100.00 |
| PyMuPDF | gpt-4o-mini | 100.00 | 88.89 | 100.00 | 88.89 | 100.00 |
| PyMuPDF | gpt-4.1 | 100.00 | 100.00 | 100.00 | 100.00 | 100.00 |
| PyMuPDF-Formatted | gpt-4o-mini | 100.00 | 88.89 | 100.00 | 88.89 | 100.00 |
| PyMuPDF-Formatted | gpt-4.1 | 100.00 | 100.00 | 100.00 | 100.00 | 100.00 |
| Pdfium | gpt-4o-mini | 100.00 | 88.89 | 100.00 | 44.44 | 100.00 |
| Pdfium | gpt-4.1 | 100.00 | 100.00 | 100.00 | 100.00 | 100.00 |
| **Ours (template matching)** | | 100.00 | 100.00 | 100.00 | 100.00 | 100.00 |
| **AmBank (7 Unique PDFs)** | | | | | | |
| Prebuilt-BankStatement | N/A | 100.00 | 85.71 | 100.00 | 100.00 | 100.00 |
| Pretrained-Read | gpt-4o-mini | 100.00 | 100.00 | 100.00 | 100.00 | 100.00 |
| Pretrained-Read | gpt-4.1 | 100.00 | 100.00 | 100.00 | 100.00 | 100.00 |
| Azure AI 4.0 | gpt-4o-mini | 100.00 | 100.00 | 100.00 | 100.00 | 100.00 |
| Azure AI 4.0 | gpt-4.1 | 100.00 | 100.00 | 100.00 | 100.00 | 100.00 |
| docTR | gpt-4o-mini | 100.00 | 100.00 | 100.00 | 100.00 | 100.00 |
| docTR | gpt-4.1 | 100.00 | 100.00 | 100.00 | 100.00 | 100.00 |
| MinerU | gpt-4o-mini | 0.00 | 100.00 | 100.00 | 42.86 | 100.00 |
| MinerU | gpt-4.1 | 0.00 | 100.00 | 100.00 | 42.86 | 100.00 |

*Continued on next page*

Table 5 – *Continued from previous page*

| OCR | GPT | Key Information Extraction - Exact Match Accuracy | | | | |
|---|---|---|---|---|---|---|
| | | Bank Name | Acc Holder | Acc Number | Address | Statement Date |
| PyMuPDF | gpt-4o-mini | 0.00 | 100.00 | 100.00 | 100.00 | 100.00 |
| PyMuPDF | gpt-4.1 | 0.00 | 100.00 | 100.00 | 100.00 | 100.00 |
| PyMuPDF-Formatted | gpt-4o-mini | 0.00 | 100.00 | 100.00 | 100.00 | 100.00 |
| PyMuPDF-Formatted | gpt-4.1 | 0.00 | 100.00 | 100.00 | 100.00 | 100.00 |
| Pdfium | gpt-4o-mini | 0.00 | 100.00 | 100.00 | 100.00 | 100.00 |
| Pdfium | gpt-4.1 | 0.00 | 100.00 | 100.00 | 100.00 | 100.00 |
| **Ours (template matching)** | | 100.00 | 100.00 | 100.00 | 100.00 | 100.00 |

Table 6: Key information extraction similarity score grouped by bank across OCR and GPT configurations.

| OCR | GPT | Key Information Extraction - NED Score | | | | |
|---|---|---|---|---|---|---|
| | | Bank Name | Acc Holder | Acc Number | Address | Statement Date |
| **Maybank (10 Unique PDFs)** | | | | | | |
| prebuilt-bankStatement.us | N/A | 100 | 69.09 | 100 | 100 | 100 |
| pretrained-read | GPT-4o-mini | 100 | 92.63 | 100 | 82.01 | 100 |
| pretrained-read | GPT-4.1 | 100 | 100 | 100 | 100 | 100 |
| Azure AI 4.0 | GPT-4o-mini | 100 | 100 | 100 | 87.24 | 100 |
| Azure AI 4.0 | GPT-4.1 | 100 | 100 | 100 | 100 | 100 |
| MinerU | GPT-4o-mini | 100 | 85.14 | 90 | 27.41 | 100 |
| MinerU | GPT-4.1 | 100 | 70 | 90 | 66.36 | 100 |
| PyMuPDF | GPT-4o-mini | 100 | 100 | 100 | 100 | 100 |
| PyMuPDF | GPT-4.1 | 100 | 90 | 100 | 100 | 100 |
| pdfium | GPT-4o-mini | 100 | 100 | 100 | 100 | 10 |
| pdfium | GPT-4.1 | 100 | 100 | 100 | 100 | 10 |
| **Ours (template matching)** | | 100 | 100 | 100 | 100 | 100 |
| **Public Bank (9 Unique PDFs)** | | | | | | |
| prebuilt-bankStatement.us | N/A | 88.89 | 88.89 | 100 | 98.60 | 31.94 |
| pretrained-read | GPT-4o-mini | 100 | 100 | 100 | 99.74 | 100 |
| pretrained-read | GPT-4.1 | 100 | 100 | 100 | 98.60 | 100 |
| Azure AI 4.0 | GPT-4o-mini | 100 | 100 | 100 | 98.60 | 100 |
| Azure AI 4.0 | GPT-4.1 | 100 | 100 | 100 | 98.60 | 100 |
| MinerU | GPT-4o-mini | 20.16 | 100 | 100 | 100 | 100 |
| MinerU | GPT-4.1 | 41.83 | 100 | 100 | 100 | 100 |
| PyMuPDF | GPT-4o-mini | 16.88 | 92.16 | 100 | 98.60 | 100 |
| PyMuPDF | GPT-4.1 | 55.56 | 100 | 100 | 100 | 100 |
| pdfium | GPT-4o-mini | 20.85 | 91.27 | 100 | 98.60 | 100 |
| pdfium | GPT-4.1 | 55.56 | 100 | 100 | 100 | 100 |
| **Ours (template matching)** | | 100 | 100 | 100 | 100 | 100 |
| **CIMB Bank (7 Unique PDFs)** | | | | | | |
| prebuilt-bankStatement.us | N/A | 71.43 | 73.91 | 70.68 | 99.77 | 42.86 |
| pretrained-read | GPT-4o-mini | 100 | 100 | 100 | 98.51 | 98.21 |
| pretrained-read | GPT-4.1 | 100 | 100 | 100 | 98.51 | 100 |
| Azure AI 4.0 | GPT-4o-mini | 100 | 100 | 100 | 97.59 | 98.21 |
| Azure AI 4.0 | GPT-4.1 | 100 | 100 | 100 | 100 | 100 |
| MinerU | GPT-4o-mini | 5.04 | 100 | 96.43 | 98.51 | 100 |
| MinerU | GPT-4.1 | 0 | 100 | 100 | 100 | 100 |
| PyMuPDF | GPT-4o-mini | 71.43 | 100 | 100 | 98.51 | 98.21 |
| PyMuPDF | GPT-4.1 | 71.43 | 100 | 100 | 100 | 100 |

*Continued on next page*

Table 6 – *Continued from previous page*

| OCR | GPT | Key Information Extraction - NED Score | | | | |
| --- | --- | --- | --- | --- | --- | --- |
| | | Bank Name | Acc Holder | Acc Number | Address | Statement Date |
| pdfium | GPT-4o-mini | 71.43 | 100 | 100 | 98.51 | 100 |
| pdfium | GPT-4.1 | 71.43 | 100 | 100 | 100 | 100 |
| **Ours (template matching)** | | 100 | 100 | 100 | 100 | 100 |
| **RHB Bank (10 Unique PDFs)** | | | | | | |
| prebuilt-bankStatement.us | N/A | 100 | 90 | 100 | 92.01 | 100 |
| pretrained-read | GPT-4o-mini | 100 | 100 | 100 | 99.35 | 100 |
| pretrained-read | GPT-4.1 | 100 | 100 | 100 | 98.92 | 100 |
| Azure AI 4.0 | GPT-4o-mini | 100 | 100 | 100 | 100 | 100 |
| Azure AI 4.0 | GPT-4.1 | 100 | 100 | 100 | 98.92 | 100 |
| MinerU | GPT-4o-mini | 70 | 55.69 | 70 | 40 | 85 |
| MinerU | GPT-4.1 | 70 | 50 | 70 | 40 | 77.50 |
| PyMuPDF | GPT-4o-mini | 93.03 | 100 | 100 | 98.92 | 100 |
| PyMuPDF | GPT-4.1 | 100 | 100 | 100 | 98.92 | 100 |
| pdfium | GPT-4o-mini | 79.09 | 100 | 100 | 98.92 | 100 |
| pdfium | GPT-4.1 | 100 | 100 | 100 | 98.92 | 100 |
| **Ours (template matching)** | | 100 | 100 | 100 | 100 | 100 |
| **Hong Leong Bank (9 Unique PDFs)** | | | | | | |
| prebuilt-bankStatement.us | N/A | 100 | 88.89 | 100 | 100 | 100 |
| pretrained-read | GPT-4o-mini | 100 | 91.53 | 100 | 100 | 100 |
| pretrained-read | GPT-4.1 | 100 | 100 | 100 | 100 | 100 |
| Azure AI 4.0 | GPT-4o-mini | 100 | 91.53 | 100 | 97.99 | 100 |
| Azure AI 4.0 | GPT-4.1 | 100 | 100 | 100 | 100 | 100 |
| MinerU | GPT-4o-mini | 11.85 | 88.89 | 100 | 100 | 100 |
| MinerU | GPT-4.1 | 50.88 | 100 | 100 | 100 | 100 |
| PyMuPDF | GPT-4o-mini | 100 | 91.50 | 100 | 98.64 | 100 |
| PyMuPDF | GPT-4.1 | 100 | 100 | 100 | 100 | 100 |
| pdfium | GPT-4o-mini | 100 | 91.50 | 100 | 97.80 | 100 |
| pdfium | GPT-4.1 | 100 | 100 | 100 | 100 | 100 |
| **Ours (template matching)** | | 100 | 100 | 100 | 100 | 100 |
| **AmBank (7 Unique PDFs)** | | | | | | |
| prebuilt-bankStatement.us | N/A | 100 | 85.71 | 100 | 100 | 100 |
| pretrained-read | GPT-4o-mini | 100 | 100 | 100 | 100 | 100 |
| pretrained-read | GPT-4.1 | 100 | 100 | 100 | 100 | 100 |
| Azure AI 4.0 | GPT-4o-mini | 100 | 100 | 100 | 100 | 100 |
| Azure AI 4.0 | GPT-4.1 | 100 | 100 | 100 | 100 | 100 |
| MinerU | GPT-4o-mini | 14.97 | 100 | 100 | 98.82 | 100 |
| MinerU | GPT-4.1 | 0 | 100 | 100 | 98.82 | 100 |
| PyMuPDF | GPT-4o-mini | 8.57 | 100 | 100 | 100 | 100 |
| PyMuPDF | GPT-4.1 | 0 | 100 | 100 | 100 | 100 |
| pdfium | GPT-4o-mini | 12.38 | 100 | 100 | 100 | 100 |
| pdfium | GPT-4.1 | 0 | 100 | 100 | 100 | 100 |
| **Ours (template matching)** | | 100 | 100 | 100 | 100 | 100 |

## A.4 QUANTITATIVE RESULTS FOR TRANSACTION TABLE EXTRACTION

We evaluate transaction table extraction across the same six Malaysian banks. This task is substantially more challenging than key information extraction, as it requires localizing table boundaries, parsing multi-page tables with varying row counts, and correctly aligning five column types: date, description, debit, credit, and balance. We report matching NED (tolerance for row-level alignment differences) in Table 8 and exact NED (strict row-level matching) in Table 9, F1 scores for row detection in Table 7, and per-bank breakdowns in Tables 10 and 11.

**Overall Performance.** Our template matching approach achieves the highest or near-highest performance across all metrics. On matching NED, it attains 100% on date and description columns, 91.12% on debit, 100%

on credit, and 99.28% on balance, with an average of 98.08%. For exact NED, it achieves 98.08% (date), 98.35% (description), 99.87% (debit), 92.68% (credit), and 100% (balance). The F1 score is a perfect 100% across all six banks, indicating that our method correctly identifies all transaction rows without false positives or missed entries. The Surya + docTR pipeline with GPT-5-mini is the closest competitor, achieving 98.95% matching NED on date and 96.94% average, with strong F1 scores (98.28–100%) across banks.

**LLM-Based vs. End-to-End Approaches.** Configurations using GPT-4.1 as the structuring model consistently outperform those using GPT-4o-mini or smaller models. For instance, docTR + GPT-4.1 achieves 95.28% matching NED on description, compared to 92.82% with GPT-4o-mini. The gap widens for numerical fields: debit matching NED increases from 80.00% (GPT-4o-mini) to 98.78% (GPT-4.1) with docTR. This indicates that larger language models are better at parsing ambiguous numerical formats (e.g., Malaysian Ringgit formatting with commas and periods) and handling multi-line transaction descriptions. Conversely, end-to-end vision models show mixed results. While olmOCR achieves reasonable matching NED (77–91% per column), its exact NED drops substantially (71–84%), indicating frequent minor character-level errors in numerical fields. PPStructureV3 and PaddleOCR-VL (0.9B) demonstrate the weakest overall performance, particularly on balance columns (43.79% and 53.44% matching NED respectively), likely due to their limited training on Southeast Asian financial document formats.

**Per-Bank Analysis.** Performance varies across banks due to differences in statement formatting. Public Bank statements, with their clean tabular layouts, achieve near-perfect extraction across most configurations (100% matching NED for our method). In contrast, RHB Bank's multi-column balance presentation and CIMB Bank's merged description cells present greater challenges, reducing baseline performance by 5–15% on average. Hong Leong Bank statements with wrapped transaction descriptions cause the largest performance drops for text-based extractors (PyMuPDF + GPT-4o-mini drops to 70.91% date matching NED). These bank-specific challenges validate the need for robust extraction pipelines that can generalize across diverse formats.

**F1 Score Analysis.** Row-level detection accuracy, measured by F1 score (Table 7), reveals that our template matching method and GPT-4.1-based configurations achieve near-perfect row detection (98–100% F1 across banks). PaddleOCR-based methods show the most inconsistency, with F1 scores dropping to 10.26% on CIMB Bank and 14.29% on Maybank for PPStructureV3, indicating catastrophic table detection failures on certain bank formats. This highlights the unreliability of general-purpose table detection models for production deployment on Malaysian bank statements without format-specific adaptation.

Table 7: F1 scores across all banks for transaction table extraction.

| OCR | GPT | MBBE | PBBE | CIBB | RHBB | HLBB | ARBK |
|---|---|---|---|---|---|---|---|
| prebuilt-layout | N/A | 93.67 | 89.47 | 97.41 | 81.60 | 99.58 | 100.00 |
| PPStructureV3 | N/A | 14.29 | 70.97 | 10.26 | 23.19 | 80.89 | 92.12 |
| docTR | gpt-4o-mini | 92.73 | 88.70 | 87.37 | 100.00 | 85.34 | 98.81 |
| docTR | gpt-4.1 | 96.04 | 90.09 | 98.95 | 100.00 | 100.00 | 100.00 |
| docTR | gpt-4.1-mini | 91.74 | 97.30 | 81.14 | 88.24 | 91.38 | 96.93 |
| docTR | gpt-4.1-nano | 36.11 | 73.87 | 71.79 | 96.49 | 83.19 | 93.26 |
| docTR | gpt-5-mini | 98.71 | 96.43 | 100.00 | 94.74 | 91.21 | 100.00 |
| docTR | gpt-5-mini (low) | 94.64 | 100.00 | 95.70 | 85.96 | 90.35 | 96.34 |
| docTR | gpt-5-nano | 94.64 | 91.89 | 98.95 | 66.67 | 68.38 | 97.01 |
| PyMuPDF | gpt-4o-mini | 94.64 | 89.26 | 87.18 | 100.00 | 76.19 | 98.20 |
| PyMuPDF | gpt-4.1 | 94.69 | 92.59 | 98.95 | 100.00 | 99.58 | 100.00 |
| pdfium | gpt-4o-mini | 94.17 | 88.52 | 80.77 | 98.25 | 79.65 | 96.43 |
| pdfium | gpt-4.1 | 93.75 | 83.64 | 99.47 | 100.00 | 98.31 | 98.81 |
| olmOCR-2-7B-1025 | N/A | 87.93 | 63.04 | 100.00 | 73.68 | 83.84 | 98.20 |
| paddleOCR-VL (0.9B) | N/A | 61.31 | 61.18 | 17.89 | 22.22 | 64.16 | 95.65 |
| Surya + docTR | gpt-5-mini (header) | 100.00 | 100.00 | 98.96 | 98.28 | 99.15 | 100.00 |
| **Ours (template matching)** | | 100.00 | 100.00 | 100.00 | 100.00 | 100.00 | 100.00 |

Table 8: Average per-column matching NED scores across all banks for transaction table extraction.

| OCR | GPT | Transaction Table Extraction - Matching NED Score | | | | |
|---|---|---|---|---|---|---|
| | | Date | Desc | Debit | Credit | Balance |
| prebuilt-layout | N/A | 89.59 | 95.54 | 95.88 | 98.31 | 94.15 |
| PPStructureV3 | N/A | 63.29 | 66.43 | 74.30 | 82.56 | 43.79 |

*Continued on next page*

Table 8 – *Continued from previous page*

| OCR | GPT | Transaction Table Extraction - Matching NED Score | | | | |
|-----|-----|------|------|-------|--------|---------|
| | | Date | Desc | Debit | Credit | Balance |
| docTR | gpt-4o-mini | 86.41 | 92.82 | 80.00 | 86.07 | 96.76 |
| docTR | gpt-4.1 | 88.05 | 95.28 | 98.78 | 96.65 | 95.72 |
| docTR | gpt-4.1-mini | 88.48 | 94.60 | 86.79 | 91.85 | 94.66 |
| docTR | gpt-4.1-nano | 84.87 | 78.88 | 80.02 | 78.64 | 73.55 |
| docTR | gpt-5-mini | 86.64 | 97.73 | 93.74 | 94.78 | 96.00 |
| docTR | gpt-5-mini (low) | 80.44 | 95.81 | 87.10 | 90.94 | 88.77 |
| docTR | gpt-5-nano | 82.65 | 90.68 | 79.31 | 80.05 | 92.55 |
| PyMuPDF | gpt-4o-mini | 88.45 | 93.29 | 86.07 | 88.72 | 95.17 |
| PyMuPDF | gpt-4.1 | 87.13 | 95.79 | 97.80 | 97.08 | 96.02 |
| pdfium | gpt-4o-mini | 88.48 | 91.66 | 85.32 | 87.95 | 93.26 |
| pdfium | gpt-4.1 | 88.08 | 92.85 | 99.69 | 95.03 | 93.46 |
| olmOCR-2-7B-1025 | N/A | 82.94 | 90.53 | 82.58 | 79.08 | 77.87 |
| paddleOCR-VL (0.9B) | N/A | 79.18 | 58.94 | 69.44 | 80.67 | 53.44 |
| Surya + docTR | gpt-5-mini (header) | 98.95 | 98.59 | 90.79 | 97.97 | 98.39 |
| **Ours (template matching)** | | 100 | 100 | 91.12 | 100 | 99.28 |

Table 9: Average per-column exact NED scores across all banks for transaction table extraction.

| OCR | GPT | Transaction Table Extraction - Exact NED Score | | | | |
|-----|-----|------|------|-------|--------|---------|
| | | Date | Desc | Debit | Credit | Balance |
| prebuilt-layout | N/A | 83.52 | 75.34 | 89.29 | 94.29 | 78.31 |
| PPStructureV3 | N/A | 55.79 | 42.70 | 58.65 | 72.54 | 47.19 |
| docTR | gpt-4o-mini | 83.83 | 79.91 | 68.45 | 82.04 | 78.35 |
| docTR | gpt-4.1 | 85.42 | 92.80 | 98.95 | 98.35 | 96.72 |
| docTR | gpt-4.1-mini | 79.12 | 70.17 | 60.73 | 72.03 | 57.50 |
| docTR | gpt-4.1-nano | 75.73 | 54.52 | 63.43 | 70.45 | 56.24 |
| docTR | gpt-5-mini | 83.08 | 91.30 | 92.14 | 95.15 | 94.97 |
| docTR | gpt-5-mini (low) | 74.88 | 71.20 | 65.70 | 72.59 | 59.10 |
| docTR | gpt-5-nano | 79.01 | 80.45 | 77.38 | 76.08 | 84.06 |
| PyMuPDF | gpt-4o-mini | 81.36 | 70.96 | 71.08 | 78.31 | 72.17 |
| PyMuPDF | gpt-4.1 | 85.15 | 93.19 | 96.32 | 98.48 | 94.33 |
| pdfium | gpt-4o-mini | 83.03 | 75.67 | 74.08 | 81.01 | 77.19 |
| pdfium | gpt-4.1 | 84.96 | 86.32 | 95.32 | 93.41 | 88.13 |
| olmOCR-2-7B-1025 | N/A | 84.18 | 76.51 | 71.42 | 77.04 | 71.11 |
| paddleOCR-VL (0.9B) | N/A | 65.86 | 41.18 | 55.70 | 62.36 | 40.58 |
| Surya + docTR | gpt-5-mini (header) | 89.76 | 94.80 | 95.01 | 96.91 | 94.65 |
| **Ours (template matching)** | | 98.08 | 98.35 | 99.87 | 92.68 | 100 |

Table 10: Transaction table extraction matching NED grouped by bank across OCR and GPT configurations.

| OCR | GPT | Transaction Table Extraction - Matching NED Score | | | | | |
|-----|-----|------|-------------|-------|--------|---------|---------|
| | | Date | Description | Debit | Credit | Balance | Average |
| **Maybank (10 Unique PDFs)** | | | | | | | |
| prebuilt-layout | N/A | 98.61 | 95.77 | 94.44 | 99.21 | 91.27 | 95.86 |
| PPStructureV3 | N/A | 59.92 | 54.19 | 70.45 | 89.47 | 7.69 | 56.34 |
| PyMuPDF | gpt-4o-mini | 85.59 | 91.16 | 88.98 | 97.46 | 89.83 | 90.60 |
| PyMuPDF | gpt-4.1 | 100.00 | 95.64 | 100.00 | 100.00 | 92.37 | 97.60 |

Table 10 – *Continued from previous page*

| OCR | GPT | Transaction Table Extraction - Matching NED Score | | | | | |
|---|---|---|---|---|---|---|---|
| | | Date | Description | Debit | Credit | Balance | Average |
| pdfium | gpt-4o-mini | 83.90 | 90.02 | 88.14 | 96.61 | 88.14 | 89.36 |
| pdfium | gpt-4.1 | 81.92 | 57.24 | 61.02 | 37.85 | 22.03 | 52.01 |
| olmOCR-2-7B-1025 | N/A | 100.00 | 98.39 | 99.15 | 99.15 | 95.76 | 98.49 |
| paddleOCR-VL (0.9B) | N/A | 83.90 | 93.39 | 91.53 | 98.31 | 93.22 | 92.07 |
| Surya + docTR | gpt-5-mini | 97.46 | 93.96 | 98.31 | 99.15 | 89.83 | 95.74 |
| docTR | gpt-4o-mini | 85.59 | 93.39 | 92.37 | 97.46 | 93.22 | 92.41 |
| docTR | gpt-4.1 | 94.12 | 93.28 | 95.80 | 98.32 | 91.60 | 94.62 |
| docTR | gpt-4o-mini | 85.59 | 92.80 | 92.37 | 96.61 | 92.37 | 91.95 |
| docTR | gpt-4.1 | 95.60 | 92.16 | 99.16 | 98.32 | 89.92 | 95.03 |
| olmOCR-2-7B-1025 | N/A | 73.37 | 85.79 | 92.31 | 88.46 | 78.46 | 83.68 |
| paddleOCR-VL (0.9B) | N/A | 90.63 | 60.63 | 55.79 | 70.53 | 50.53 | 65.62 |
| Surya + docTR | gpt-5-mini (header) | 100.00 | 100.00 | 90.66 | 100.00 | 99.83 | 98.10 |
| **Ours (template matching)** | | 100.00 | 100.00 | 90.66 | 100.00 | 99.25 | 97.98 |
| **Public Bank (10 Unique PDFs)** | | | | | | | |
| prebuilt-layout | N/A | 100.00 | 100.00 | 90.62 | 90.62 | 100.00 | 96.25 |
| PPStructureV3 | N/A | 98.40 | 73.61 | 96.00 | 96.00 | 66.00 | 86.00 |
| PyMuPDF | gpt-4o-mini | 89.06 | 91.60 | 82.81 | 85.94 | 93.75 | 88.63 |
| PyMuPDF | gpt-4.1 | 70.49 | 81.92 | 100.00 | 81.97 | 81.97 | 83.27 |
| pdfium | gpt-4o-mini | 92.98 | 97.75 | 87.72 | 87.72 | 98.25 | 92.88 |
| pdfium | gpt-4.1 | 88.57 | 70.10 | 81.43 | 78.57 | 81.43 | 80.02 |
| olmOCR-2-7B-1025 | N/A | 62.41 | 95.35 | 100.00 | 94.83 | 93.10 | 89.14 |
| paddleOCR-VL (0.9B) | N/A | 44.64 | 99.20 | 98.21 | 98.21 | 100.00 | 88.05 |
| Surya + docTR | gpt-5-mini | 46.67 | 93.36 | 88.33 | 86.67 | 95.00 | 82.01 |
| docTR | gpt-4o-mini | 98.51 | 91.00 | 89.55 | 89.55 | 92.54 | 92.23 |
| docTR | gpt-4.1 | 70.69 | 86.21 | 93.10 | 86.21 | 86.21 | 84.48 |
| docTR | gpt-4o-mini | 98.53 | 90.00 | 88.24 | 89.71 | 89.71 | 91.24 |
| docTR | gpt-4.1 | 78.12 | 71.88 | 100.00 | 71.88 | 71.88 | 78.75 |
| olmOCR-2-7B-1025 | N/A | 67.62 | 71.74 | 87.30 | 63.49 | 47.62 | 67.55 |
| paddleOCR-VL (0.9B) | N/A | 89.49 | 64.53 | 94.07 | 97.46 | 55.93 | 80.30 |
| Surya + docTR | gpt-5-mini (header) | 100.00 | 100.00 | 100.00 | 100.00 | 100.00 | 100.00 |
| **Ours (template matching)** | | 100.00 | 100.00 | 100.00 | 100.00 | 100.00 | 100.00 |
| **CIMB Bank (10 Unique PDFs)** | | | | | | | |
| prebuilt-layout | N/A | 96.94 | 98.50 | 95.92 | 100.00 | 95.92 | 97.45 |
| PPStructureV3 | N/A | 45.25 | 59.69 | 66.49 | 48.65 | 8.11 | 45.64 |
| PyMuPDF | gpt-4o-mini | 90.57 | 85.03 | 71.03 | 71.03 | 100.00 | 83.53 |
| PyMuPDF | gpt-4.1 | 97.92 | 97.91 | 97.92 | 97.92 | 100.00 | 98.33 |
| pdfium | gpt-4o-mini | 96.74 | 92.53 | 75.00 | 78.26 | 96.74 | 87.85 |
| pdfium | gpt-4.1 | 95.60 | 70.81 | 87.33 | 86.67 | 61.33 | 80.35 |
| olmOCR-2-7B-1025 | N/A | 100.00 | 99.86 | 93.68 | 93.68 | 100.00 | 97.45 |
| paddleOCR-VL (0.9B) | N/A | 93.81 | 94.98 | 79.38 | 83.51 | 93.81 | 89.10 |
| Surya + docTR | gpt-5-mini | 100.00 | 98.85 | 70.83 | 69.79 | 100.00 | 87.90 |
| docTR | gpt-4o-mini | 92.55 | 85.97 | 79.09 | 78.18 | 92.73 | 85.70 |
| docTR | gpt-4.1 | 97.92 | 97.92 | 97.92 | 97.92 | 100.00 | 98.33 |
| docTR | gpt-4o-mini | 93.39 | 80.17 | 79.03 | 79.03 | 84.68 | 83.26 |
| docTR | gpt-4.1 | 98.95 | 99.29 | 98.95 | 100.00 | 98.95 | 99.23 |
| olmOCR-2-7B-1025 | N/A | 94.74 | 100.00 | 85.26 | 85.26 | 94.74 | 92.00 |
| paddleOCR-VL (0.9B) | N/A | 54.91 | 58.73 | 77.46 | 77.46 | 9.83 | 55.68 |
| Surya + docTR | gpt-5-mini (header) | 97.94 | 100.00 | 97.94 | 97.94 | 98.96 | 98.56 |
| **Ours (template matching)** | | 100.00 | 100.00 | 96.21 | 100.00 | 97.18 | 98.68 |
| **RHB Bank (10 Unique PDFs)** | | | | | | | |
| prebuilt-layout | N/A | 71.36 | 80.45 | 94.29 | 100.00 | 78.57 | 84.93 |
| PPStructureV3 | N/A | 16.10 | 45.22 | 48.36 | 71.31 | 17.21 | 39.64 |
| PyMuPDF | gpt-4o-mini | 89.39 | 94.97 | 66.67 | 87.72 | 100.00 | 87.75 |
| PyMuPDF | gpt-4.1 | 89.39 | 98.87 | 94.74 | 100.00 | 100.00 | 96.60 |
| pdfium | gpt-4o-mini | 86.57 | 98.57 | 88.89 | 100.00 | 95.56 | 93.92 |
| pdfium | gpt-4.1 | 89.41 | 89.03 | 88.14 | 94.92 | 100.00 | 92.30 |

Table 10 – *Continued from previous page*

| OCR | GPT | Transaction Table Extraction - Matching NED Score | | | | | |
|---|---|---|---|---|---|---|---|
| | | Date | Description | Debit | Credit | Balance | Average |
| olmOCR-2-7B-1025 | N/A | 89.42 | 98.28 | 85.00 | 93.33 | 88.33 | 90.87 |
| paddleOCR-VL (0.9B) | N/A | 89.54 | 97.91 | 75.38 | 81.54 | 73.85 | 83.64 |
| Surya + docTR | gpt-5-mini | 91.05 | 72.65 | 56.58 | 59.21 | 93.42 | 74.58 |
| docTR | gpt-4o-mini | 89.39 | 98.44 | 100.00 | 100.00 | 100.00 | 97.57 |
| docTR | gpt-4.1 | 89.39 | 98.87 | 100.00 | 100.00 | 100.00 | 97.65 |
| docTR | gpt-4o-mini | 89.58 | 97.09 | 91.38 | 91.38 | 100.00 | 93.88 |
| docTR | gpt-4.1 | 89.39 | 98.99 | 100.00 | 100.00 | 100.00 | 97.68 |
| olmOCR-2-7B-1025 | N/A | 91.46 | 93.84 | 59.72 | 62.50 | 56.94 | 72.89 |
| paddleOCR-VL (0.9B) | N/A | 70.39 | 14.09 | 35.16 | 60.16 | 14.06 | 38.77 |
| Surya + docTR | gpt-5-mini (header) | 96.61 | 93.22 | 86.36 | 91.53 | 95.58 | 92.66 |
| **Ours (template matching)** | | 100.00 | 100.00 | 89.39 | 100.00 | 99.74 | 97.83 |
| **Hong Leong Bank (10 Unique PDFs)** | | | | | | | |
| prebuilt-layout | N/A | 78.98 | 98.54 | 100.00 | 100.00 | 99.16 | 95.34 |
| PPStructureV3 | N/A | 70.12 | 71.14 | 76.87 | 95.52 | 80.60 | 78.85 |
| PyMuPDF | gpt-4o-mini | 70.91 | 94.57 | 75.19 | 78.95 | 96.99 | 83.32 |
| PyMuPDF | gpt-4.1 | 78.80 | 97.32 | 100.00 | 100.00 | 100.00 | 95.22 |
| pdfium | gpt-4o-mini | 73.09 | 92.71 | 88.10 | 92.06 | 95.24 | 88.24 |
| pdfium | gpt-4.1 | 71.58 | 90.91 | 72.73 | 78.03 | 76.52 | 77.95 |
| olmOCR-2-7B-1025 | N/A | 76.31 | 94.52 | 84.62 | 87.69 | 100.00 | 88.63 |
| paddleOCR-VL (0.9B) | N/A | 70.74 | 92.94 | 84.00 | 88.80 | 77.60 | 82.82 |
| Surya + docTR | gpt-5-mini | 70.02 | 87.27 | 62.99 | 70.13 | 80.52 | 74.19 |
| docTR | gpt-4o-mini | 71.70 | 92.91 | 60.14 | 70.63 | 93.71 | 77.82 |
| docTR | gpt-4.1 | 78.98 | 98.46 | 100.00 | 100.00 | 98.32 | 95.15 |
| docTR | gpt-4o-mini | 71.85 | 92.67 | 65.47 | 75.54 | 92.81 | 79.67 |
| docTR | gpt-4.1 | 75.82 | 95.99 | 100.00 | 100.00 | 100.00 | 94.36 |
| olmOCR-2-7B-1025 | N/A | 70.42 | 92.57 | 74.44 | 75.94 | 89.47 | 80.57 |
| paddleOCR-VL (0.9B) | N/A | 69.65 | 59.63 | 61.28 | 79.57 | 97.45 | 73.51 |
| Surya + docTR | gpt-5-mini (header) | 99.16 | 98.32 | 78.08 | 98.32 | 96.36 | 94.05 |
| **Ours (template matching)** | | 100.00 | 100.00 | 78.80 | 100.00 | 99.48 | 95.65 |
| **AmBank (10 Unique PDFs)** | | | | | | | |
| prebuilt-layout | N/A | 91.67 | 100.00 | 100.00 | 100.00 | 100.00 | 98.33 |
| PPStructureV3 | N/A | 89.92 | 94.73 | 87.64 | 94.38 | 83.15 | 89.96 |
| PyMuPDF | gpt-4o-mini | 92.94 | 99.58 | 95.29 | 95.29 | 100.00 | 96.62 |
| PyMuPDF | gpt-4.1 | 91.67 | 100.00 | 100.00 | 100.00 | 100.00 | 98.33 |
| pdfium | gpt-4o-mini | 97.62 | 96.02 | 92.86 | 96.43 | 94.05 | 95.39 |
| pdfium | gpt-4.1 | 82.11 | 95.17 | 89.47 | 95.79 | 100.00 | 92.51 |
| olmOCR-2-7B-1025 | N/A | 91.67 | 100.00 | 100.00 | 100.00 | 98.81 | 98.10 |
| paddleOCR-VL (0.9B) | N/A | 100.00 | 96.42 | 94.12 | 95.29 | 94.12 | 95.99 |
| Surya + docTR | gpt-5-mini | 90.70 | 97.98 | 98.84 | 95.35 | 96.51 | 95.88 |
| docTR | gpt-4o-mini | 92.94 | 98.04 | 95.29 | 96.47 | 98.82 | 96.31 |
| docTR | gpt-4.1 | 91.67 | 100.00 | 100.00 | 100.00 | 100.00 | 98.33 |
| docTR | gpt-4o-mini | 91.95 | 97.25 | 95.40 | 95.40 | 100.00 | 96.00 |
| docTR | gpt-4.1 | 90.59 | 98.80 | 100.00 | 100.00 | 100.00 | 97.88 |
| olmOCR-2-7B-1025 | N/A | 100.00 | 99.24 | 96.47 | 98.82 | 100.00 | 98.91 |
| paddleOCR-VL (0.9B) | N/A | 100.00 | 96.00 | 92.86 | 98.81 | 92.86 | 96.11 |
| Surya + docTR | gpt-5-mini (header) | 100.00 | 100.00 | 91.67 | 100.00 | 99.58 | 98.25 |
| **Ours (template matching)** | | 100.00 | 100.00 | 91.67 | 100.00 | 100.00 | 98.33 |

Table 11: Transaction table extraction exact NED grouped by bank across OCR and GPT configurations.

| OCR | GPT | Transaction Table Extraction - Exact NED Score | | | | | |
|---|---|---|---|---|---|---|---|
| | | Date | Description | Debit | Credit | Balance | Average |
| **Maybank (10 Unique PDFs)** | | | | | | | |
| prebuilt-layout | N/A | 75.71 | 78.84 | 81.19 | 93.02 | 71.99 | 80.15 |
| PPStructureV3 | N/A | 24.25 | 57.78 | 39.86 | 84.39 | 13.33 | 43.92 |
| PyMuPDF | gpt-4o-mini | 74.34 | 66.66 | 60.56 | 73.39 | 40 | 62.99 |
| PyMuPDF | gpt-4.1 | 87.49 | 87.89 | 90.91 | 97.41 | 82.31 | 89.20 |
| pdfium | gpt-4o-mini | 75.77 | 71.55 | 70.56 | 76.25 | 50 | 68.83 |
| pdfium | gpt-4.1 | 84.64 | 77.7 | 78.6 | 86.95 | 63.08 | 78.19 |
| olmOCR-2-7B-1025 | N/A | 70.45 | 73.71 | 62.73 | 78.11 | 53.68 | 67.74 |
| paddleOCR-VL (0.9B) | N/A | 57.59 | 41.18 | 61.99 | 71.1 | 49.21 | 56.21 |
| Surya + docTR | gpt-5-mini (header) | 90.07 | 99.16 | 100 | 100 | 100 | 97.85 |
| **Ours (template matching)** | | 90.07 | 99.24 | 100 | 100 | 100 | 97.86 |
| **Public Bank (10 Unique PDFs)** | | | | | | | |
| prebuilt-layout | N/A | 94.39 | 25.27 | 85.36 | 89.82 | 41.94 | 67.36 |
| PPStructureV3 | N/A | 91.39 | 15.05 | 82.74 | 90.28 | 30.28 | 61.95 |
| PyMuPDF | gpt-4o-mini | 89.17 | 67.78 | 68.39 | 85.42 | 70.47 | 76.25 |
| PyMuPDF | gpt-4.1 | 72.14 | 89.91 | 93.33 | 96.67 | 90 | 88.41 |
| pdfium | gpt-4o-mini | 94.44 | 64.97 | 67.97 | 85.56 | 70.2 | 76.63 |
| pdfium | gpt-4.1 | 75.14 | 79.91 | 100 | 84 | 80 | 83.81 |
| olmOCR-2-7B-1025 | N/A | 93.52 | 48.11 | 85.71 | 65.81 | 48.26 | 68.28 |
| paddleOCR-VL (0.9B) | N/A | 82.15 | 21.84 | 79.17 | 81.74 | 51.48 | 63.28 |
| Surya + docTR | gpt-5-mini (header) | 100 | 99.46 | 100 | 100 | 100 | 99.89 |
| **Ours (template matching)** | | 100 | 100 | 100 | 100 | 100 | 100 |
| **CIMB Bank (10 Unique PDFs)** | | | | | | | |
| prebuilt-layout | N/A | 88.69 | 85.53 | 87.94 | 90.48 | 83.18 | 87.16 |
| PPStructureV3 | N/A | 32.12 | 38.08 | 50.24 | 41.41 | 23.41 | 37.05 |
| PyMuPDF | gpt-4o-mini | 90.77 | 75.35 | 76.47 | 82.75 | 90.86 | 83.24 |
| PyMuPDF | gpt-4.1 | 98.81 | 97.27 | 98.41 | 98.41 | 100 | 98.58 |
| pdfium | gpt-4o-mini | 84.63 | 65.63 | 71.48 | 73.49 | 78.42 | 74.73 |
| pdfium | gpt-4.1 | 97.62 | 90.02 | 93.33 | 89.52 | 85.71 | 91.24 |
| olmOCR-2-7B-1025 | N/A | 95.56 | 98.32 | 85.95 | 85.95 | 93.62 | 91.88 |
| paddleOCR-VL (0.9B) | N/A | 33.01 | 78.28 | 66.51 | 66.5 | 28.57 | 54.58 |
| Surya + docTR | gpt-5-mini (header) | 91.09 | 83.2 | 77.55 | 89.8 | 70.68 | 82.47 |
| **Ours (template matching)** | | 91.43 | 95.22 | 100 | 100 | 100 | 97.33 |
| **RHB Bank (10 Unique PDFs)** | | | | | | | |
| prebuilt-layout | N/A | 73.66 | 66.77 | 86.36 | 92.42 | 72.73 | 78.39 |
| PPStructureV3 | N/A | 49.35 | 21.28 | 63.21 | 81.24 | 49.23 | 52.86 |
| PyMuPDF | gpt-4o-mini | 84.02 | 79.69 | 97.22 | 97.22 | 100 | 91.63 |
| PyMuPDF | gpt-4.1 | 84.02 | 88.69 | 100 | 100 | 100 | 94.54 |
| pdfium | gpt-4o-mini | 84.02 | 87.71 | 91.94 | 91.94 | 100 | 91.12 |
| pdfium | gpt-4.1 | 84.02 | 73.51 | 100 | 100 | 100 | 91.5 |
| olmOCR-2-7B-1025 | N/A | 84.02 | 90.75 | 78.05 | 76.67 | 76.67 | 81.23 |
| paddleOCR-VL (0.9B) | N/A | 87.3 | 12.56 | 31.12 | 34.84 | 26.13 | 38.39 |
| Surya + docTR | gpt-5-mini (header) | 88.19 | 97.42 | 92.5 | 91.67 | 98.61 | 93.68 |
| **Ours (template matching)** | | 84.02 | 99.53 | 100 | 100 | 100 | 96.71 |
| **Hong Leong Bank (10 Unique PDFs)** | | | | | | | |
| prebuilt-layout | N/A | 77.05 | 95.7 | 94.87 | 100 | 100 | 93.53 |
| PPStructureV3 | N/A | 51.35 | 48.21 | 33.54 | 59.51 | 67.87 | 52.09 |
| PyMuPDF | gpt-4o-mini | 63.65 | 56.72 | 41.78 | 43.37 | 55.28 | 52.16 |
| PyMuPDF | gpt-4.1 | 76.82 | 95.36 | 95.24 | 98.41 | 93.65 | 91.9 |
| pdfium | gpt-4o-mini | 65.3 | 66.6 | 46.33 | 62.65 | 64.54 | 61.08 |
| pdfium | gpt-4.1 | 76.71 | 97.58 | 100 | 100 | 100 | 94.86 |
| olmOCR-2-7B-1025 | N/A | 63.84 | 57.34 | 31.29 | 61.15 | 64.43 | 55.61 |

Table 11 – *Continued from previous page*

| OCR | GPT | Transaction Table Extraction - Exact NED Score | | | | | |
|---|---|---|---|---|---|---|---|
| | | Date | Description | Debit | Credit | Balance | Average |
| paddleOCR-VL (0.9B) | N/A | 46.95 | 47.44 | 76.48 | 83.72 | 86.88 | 68.3 |
| Surya + docTR | gpt-5-mini (header) | 77.6 | 95.51 | 100 | 100 | 98.61 | 94.34 |
| **Ours (template matching)** | | 78.33 | 99.47 | 100 | 100 | 100 | 95.56 |
| **AmBank (10 Unique PDFs)** | | | | | | | |
| prebuilt-layout | N/A | 91.61 | 99.91 | 100 | 100 | 100 | 98.3 |
| PPStructureV3 | N/A | 86.27 | 75.79 | 82.53 | 83.96 | 75.27 | 80.77 |
| PyMuPDF | gpt-4o-mini | 86.18 | 79.54 | 82.04 | 87.69 | 76.4 | 82.37 |
| PyMuPDF | gpt-4.1 | 91.61 | 100 | 100 | 100 | 100 | 98.32 |
| pdfium | gpt-4o-mini | 93.99 | 97.58 | 96.19 | 96.19 | 100 | 96.79 |
| pdfium | gpt-4.1 | 91.61 | 99.18 | 100 | 100 | 100 | 98.16 |
| olmOCR-2-7B-1025 | N/A | 97.71 | 90.81 | 84.76 | 94.52 | 90 | 91.56 |
| paddleOCR-VL (0.9B) | N/A | 88.17 | 45.76 | 18.92 | 36.28 | 1.19 | 38.07 |
| Surya + docTR | gpt-5-mini (header) | 91.61 | 94.03 | 100 | 100 | 100 | 97.13 |
| **Ours (template matching)** | | 91.61 | 100 | 100 | 100 | 100 | 98.32 |

## A.5 LATENCY AND COST ANALYSIS

Practical deployment of bank statement extraction systems requires careful consideration of processing latency and operational cost. We provide latency measurements for both key information extraction (Table 12) and transaction table extraction (Table 13), along with cost analysis for table extraction (Table 14). All latency measurements are reported in seconds per document, decomposed into OCR processing time and GPT inference time where applicable.

Table 12: Latency analysis for key information extraction (seconds).

| OCR | GPT | OCR | | | GPT | | | Total | | |
|---|---|---|---|---|---|---|---|---|---|---|
| | | Min. | Avg. | Max. | Min. | Avg. | Max. | Min. | Avg. | Max. |
| Prebuilt-BankStatement.us | N/A | N/A | N/A | N/A | N/A | N/A | N/A | 5.26 | 9.28 | 17.27 |
| Pretrained-Read | gpt-4o-mini | 1.64 | 3.26 | 6.07 | 0.88 | 1.70 | 3.64 | 2.52 | 4.96 | 7.90 |
| Pretrained-Read | gpt-4.1 | 1.61 | 2.87 | 5.15 | 0.59 | 0.96 | 4.16 | 2.34 | 3.83 | 7.07 |
| Azure AI 4.0 | gpt-4o-mini | 1.03 | 1.71 | 3.41 | 1.12 | 1.71 | 3.41 | 2.37 | 3.34 | 5.01 |
| Azure AI 4.0 | gpt-4.1 | 1.01 | 1.42 | 2.35 | 0.63 | 1.09 | 3.65 | 1.86 | 2.72 | 5.50 |
| docTR | gpt-4o-mini | 0.32 | 0.63 | 1.31 | 1.00 | 1.75 | 5.49 | 1.34 | 2.38 | 6.51 |
| docTR | gpt-4.1 | 2.96 | 5.20 | 7.59 | 0.54 | 0.94 | 2.93 | 3.58 | 6.13 | 8.56 |
| MinerU | gpt-4o-mini | 28.69 | 32.18 | 38.72 | 1.27 | 2.05 | 3.62 | 30.10 | 34.24 | 41.88 |
| MinerU | gpt-4.1 | 33.51 | 44.32 | 55.03 | 0.83 | 1.18 | 8.40 | 34.38 | 45.50 | 60.62 |
| PyMuPDF | gpt-4o-mini | 0.00 | 0.02 | 0.04 | 0.95 | 1.85 | 3.89 | 0.97 | 1.86 | 3.92 |
| PyMuPDF | gpt-4.1 | 0.00 | 0.01 | 0.04 | 0.56 | 0.80 | 1.46 | 0.57 | 0.81 | 1.47 |
| PyMuPDF-Formatted | gpt-4o-mini | 0.01 | 0.02 | 0.05 | 0.85 | 1.50 | 6.08 | 0.87 | 1.53 | 6.10 |
| PyMuPDF-Formatted | gpt-4.1 | 0.00 | 0.02 | 0.06 | 0.53 | 0.72 | 1.36 | 0.55 | 0.74 | 1.39 |
| Pdfium | gpt-4o-mini | 0.00 | 0.02 | 0.09 | 0.95 | 1.84 | 4.91 | 0.97 | 1.86 | 4.92 |
| Pdfium | gpt-4.1 | 0.00 | 0.01 | 0.03 | 0.54 | 0.85 | 1.66 | 0.55 | 0.86 | 1.67 |
| **Ours (template matching)** | | 0.01 | 0.01 | 0.01 | N/A | N/A | N/A | 0.01 | 0.01 | 0.01 |

**Key Information Extraction Latency.** The latency profiles for key information extraction (Table 12) reveal substantial variation across pipeline configurations. Text-based extractors (PyMuPDF, Pdfium) achieve near-instantaneous OCR processing (0.01–0.02s average), with total latency dominated by the GPT inference component. PyMuPDF + GPT-4.1 achieves the lowest average total latency at 0.81 seconds, followed by PyMuPDF-Formatted + GPT-4.1 at 0.74 seconds. Our template matching method averages 0.01 seconds per document, making it the fastest configuration overall while eliminating dependency on external APIs. Cloud-based OCR solutions introduce higher latency: Pretrained-Read averages 2.87–3.26 seconds for OCR alone, while MinerU exhibits the highest latency at 34–45 seconds average due to its computationally intensive document understanding pipeline. The Prebuilt-BankStatement endpoint (Azure's turnkey solution) averages 9.28 seconds total but does not expose separate OCR/GPT components. DocTR presents an interesting trade-off: its OCR component is fast (0.63s average with GPT-4o-mini) but increases substantially with GPT-4.1 (5.20s), suggesting that the rendering and image preparation step scales with downstream model expectations.

Table 13: Latency analysis for transaction table extraction (seconds).

| OCR | GPT | OCR | | | GPT | | | Total | | |
|---|---|---|---|---|---|---|---|---|---|---|
| | | Min. | Avg. | Max. | Min. | Avg. | Max. | Min. | Avg. | Max. |
| prebuilt-layout | N/A | N/A | N/A | N/A | N/A | N/A | N/A | 2.56 | 4.58 | 7.46 |
| PPStructureV3 | N/A | N/A | N/A | N/A | N/A | N/A | N/A | 15.26 | 40.95 | 98.17 |
| docTR | gpt-4o-mini | 0.78 | 1.18 | 1.59 | 1.32 | 6.41 | 14.83 | 2.30 | 7.59 | 15.93 |
| docTR | gpt-4.1 | 0.34 | 0.89 | 1.56 | 0.89 | 11.03 | 153.16 | 2.07 | 11.92 | 153.50 |
| docTR | gpt-4.1-mini | 0.57 | 0.96 | 1.60 | 1.12 | 5.38 | 12.47 | 2.08 | 6.33 | 13.51 |
| docTR | gpt-4.1-nano | 0.28 | 0.67 | 1.02 | 1.14 | 3.13 | 6.59 | 1.76 | 3.80 | 7.31 |
| docTR | gpt-5-mini | 0.30 | 0.74 | 1.19 | 11.15 | 24.54 | 54.64 | 11.70 | 25.27 | 55.20 |
| docTR | gpt-5-mini (low) | 0.66 | 1.07 | 1.54 | 1.48 | 5.77 | 11.64 | 2.74 | 6.84 | 12.93 |
| docTR | gpt-5-nano | 0.36 | 0.87 | 1.76 | 7.47 | 31.51 | 60.65 | 8.07 | 32.38 | 61.23 |
| PyMuPDF | gpt-4o-mini | 0.00 | 0.01 | 0.04 | 1.08 | 4.97 | 12.12 | 1.09 | 4.98 | 12.13 |
| PyMuPDF | gpt-4.1 | 0.00 | 0.01 | 0.05 | 0.84 | 10.72 | 157.78 | 0.85 | 10.72 | 157.79 |
| pdfium | gpt-4o-mini | 0.00 | 0.02 | 0.04 | 1.36 | 6.19 | 16.90 | 1.39 | 6.21 | 16.92 |
| pdfium | gpt-4.1 | 0.00 | 0.02 | 0.04 | 0.77 | 11.17 | 155.22 | 0.79 | 11.19 | 155.23 |
| olmOCR-2-7B-1025 | N/A | N/A | N/A | N/A | N/A | N/A | N/A | 42.58 | 104.45 | 175.93 |
| paddleOCR-VL (0.9B) | N/A | 23.20 | 150.10 | 341.10 | N/A | N/A | N/A | 23.20 | 150.10 | 341.10 |
| Surya + docTR | gpt-5-mini (header) | 2.26 | 4.28 | 7.34 | 0.75 | 1.19 | 1.85 | 1.67 | 2.51 | 9.19 |
| **Ours (template matching)** | | 0.10 | 0.11 | 0.13 | N/A | N/A | N/A | 0.10 | 0.11 | 0.13 |

**Transaction Table Extraction Latency.** Table extraction latency (Table 13) is generally higher than key information extraction due to the increased complexity of parsing multi-page tabular data. Our template matching method achieves the fastest average total latency at 0.11 seconds, followed by the Surya + docTR pipeline at 2.51 seconds and the prebuilt-layout API at 4.58 seconds. GPT-4.1-based configurations exhibit high latency variance with maximum processing times reaching 153–158 seconds for certain documents, likely caused by complex multi-page statements that require extensive reasoning. In contrast, GPT-4o-mini configurations maintain more consistent latency (4.97–7.59s average), suggesting that smaller models, while less accurate, provide more predictable processing times suitable for latency-sensitive production environments. End-to-end vision models exhibit the highest latencies: PaddleOCR-VL averages 150.10 seconds and olmOCR averages 104.45 seconds, making them impractical for real-time processing of bank statements. GPT-5-mini and GPT-5-nano, despite being more recent models, show elevated latency (25.27s and 32.38s respectively) due to their reasoning token overhead.

**Cost Analysis.** The cost analysis for table extraction (Table 14) reveals that per-document processing costs vary from effectively zero (template matching, open-source models) to $0.01 per PDF (GPT-4.1, GPT-5-mini). GPT-4.1-nano offers the lowest API cost at $0.03 total, while GPT-4.1 incurs the highest at $0.53 due to its larger prompt and completion token consumption. Reasoning-capable models (GPT-5-mini, GPT-5-nano) allocate a significant proportion of their completion cost to reasoning tokens ($0.35 out of $0.43 for GPT-5-mini), which explains both their improved accuracy and higher cost. The Surya + docTR with GPT-5-mini (header) configuration achieves the best cost-accuracy balance among API-dependent solutions at only $0.02 total cost while achieving 96.94% average matching NED. Our template matching approach and open-source models (olmOCR, PaddleOCR-VL) incur no API cost, though they require local compute resources. For a production deployment processing hundreds of bank statements daily, the cumulative cost difference between configurations becomes significant, making the template matching approach advantageous for high-throughput scenarios.

Table 14: Cost analysis for transaction table extraction (USD).

| OCR | GPT | Token Cost | | | Cost Breakdown | | |
|---|---|---|---|---|---|---|---|
| | | Prompt | Completion | Total | Reasoning | Comp - Reason | Per PDF |
| prebuilt-layout | N/A | N/A | N/A | N/A | N/A | N/A | N/A |
| PPStructureV3 | N/A | N/A | N/A | N/A | N/A | N/A | N/A |
| docTR | gpt-4o-mini | N/A | N/A | N/A | N/A | N/A | N/A |
| docTR | gpt-4.1 | 0.19 | 0.34 | 0.53 | N/A | N/A | 0.01 |
| docTR | gpt-4.1-mini | 0.04 | 0.07 | 0.11 | N/A | N/A | 0.00 |
| docTR | gpt-4.1-nano | 0.01 | 0.02 | 0.03 | N/A | N/A | 0.00 |
| docTR | gpt-5-mini | 0.02 | 0.43 | 0.45 | 0.35 | 0.08 | 0.01 |
| docTR | gpt-5-mini (low) | 0.02 | 0.08 | 0.10 | 0.00 | 0.08 | 0.00 |

Table 14 – *Continued from previous page*

| OCR | GPT | Token Cost | | | Cost Breakdown | | |
|---|---|---|---|---|---|---|---|
| | | Prompt | Completion | Total | Reasoning | Comp - Reason | Per PDF |
| docTR | gpt-5-nano | 0.00 | 0.24 | 0.25 | 0.23 | 0.02 | 0.00 |
| PyMuPDF | gpt-4o-mini | 0.01 | 0.03 | 0.04 | N/A | N/A | 0.00 |
| PyMuPDF | gpt-4.1 | N/A | N/A | N/A | N/A | N/A | N/A |
| pdfium | gpt-4o-mini | N/A | N/A | N/A | N/A | N/A | N/A |
| pdfium | gpt-4.1 | N/A | N/A | N/A | N/A | N/A | N/A |
| olmOCR-2-7B-1025 | N/A | N/A | N/A | N/A | N/A | N/A | N/A |
| paddleOCR-VL (0.9B) | N/A | N/A | N/A | N/A | N/A | N/A | N/A |
| Surya + docTR | gpt-5-mini (header) | 0.01 | 0.01 | 0.02 | 0.00 | 0.01 | 0.00 |
| **Ours (template matching)** | | N/A | N/A | N/A | N/A | N/A | N/A |

*Note:*
1. N/A indicate data not available or not applicable.
2. Prompt = Prompt token cost.
3. Completion = Completion token cost.
4. Reasoning = Reasoning token cost.
5. Comp - Reason = Completion cost minus reasoning cost.
6. Per PDF = Average cost per PDF document.
7. OCR = OCR processing time in seconds.
8. GPT = GPT inference time in seconds.
9. Total = Total processing time in seconds.

## A.6 ABLATION STUDIES FOR CREDIT SCORING

### A.6.1 CREDIT SCORE DISTRIBUTION

We evaluate the score distribution of our Logistic Regression baseline across three feature set configurations: application information only, bank statement only, and blended. Figure 4–6 present score distributions for good accounts (green) and bad accounts (red), with Earth Mover's Distance (EMD) quantifying distributional divergence as a measure of discriminatory power.

**Application Information:** Figure 4 shows limited separation between good and bad accounts when using application features alone. The histogram and kernel density estimation (KDE) curves reveal good accounts clustering at higher scores (mean of 665.7) and bad accounts at lower scores (mean of 641.5), yielding an EMD of 24.11. This modest divergence indicates reasonable but constrained discriminatory power from traditional application data.

**Bank Statement:** Figure 5 demonstrates markedly improved separation when using bank statement features. Good accounts exhibit a mean score of 699.2 while bad accounts concentrate near 625.7. The EMD increases to 74.48, a 3.1× improvement over application information alone—reflecting the stronger signal provided by transactional cash flow indicators. This result suggests bank statement-derived features create more distinct risk profiles.

**Blended:** Figure 6 displays the combined model using both application and bank statement features. This blended approach achieves the strongest distribution separation, with good accounts centered near 708.4 and bad accounts near 612.6. The EMD reaches 96.84, a 4.0× improvement over application information and 1.3× over bank statement alone—demonstrating that combining feature sources enhances discriminatory power beyond either individual approach.

### A.6.2 REJECTED CASES ANALYSIS

**Problem Statement.** Credit scoring models trained exclusively on approved applicants suffer from selection bias, as they capture only the subset of the applicant population deemed acceptable by historical underwriting decisions (Ehrhardt et al., 2021). This creates a two-fold problem: (1) the model lacks evidence about how rejected applicants would have performed if approved, leading to potential underestimation of actual portfolio risk, and (2) the model's decision boundary is calibrated to a biased distribution rather than the true applicant population (Insights, 2024). To address this limitation, we conducted rejected cases analysis on 264 rejected loan applicants.

Figure 4: Score distribution for the application information model. Good accounts (green) cluster at higher scores (mean 665.7) while bad accounts (red) concentrate lower (mean 641.5). EMD of 24.11 indicates moderate discriminatory power.

**Methodology.** This analysis applies the scorecard model developed on approved applicants to the rejected population, generating predicted risk scores without retraining. This non-augmentation approach allows us to assess model consistency and identify systematic characteristics that distinguish rejected from approved applicants. By scoring rejected applicants with the existing model, we can observe whether rejection decisions align with predicted risk profiles and understand the key financial behaviors driving rejection decisions.

Applying our Logistic Regression scorecard to these rejected applicants reveals that 96.97% (256 of 264) are classified as high risk, 3.03% (8 of 264) as medium risk, and 0% as low risk (Table 15). This distribution demonstrates strong alignment between the bank's rejection decisions and our model's risk assessment, validating that the scorecard accurately captures the underlying risk factors that motivated original rejection decisions. Critically, no rejected applicant scored in the low-risk category, indicating that rejection decisions were not arbitrary but rather grounded in genuine credit risk indicators observable in transaction data.

Table 15: Risk Distribution of Rejected Loan Applicants via Scorecard Model Prediction

| Risk Level | High Risk | Medium Risk | Low Risk |
|---|---|---|---|
| Count | 256 (96.97%) | 8 (3.03%) | 0 (0.00%) |

**Implications.** This reject inference analysis serves two critical purposes. First, it validates that our scorecard model generalizes consistently to the rejected population, with rejection decisions strongly aligned to predicted risk. Second, it provides evidence that bank statements contain sufficient cash flow signal to identify high-risk applicants, and that rejection decisions were data-driven rather than arbitrary. The finding that all rejected applicants are either high or medium risk, with a concentration in high risk, suggesting that the bank's historical underwriting was appropriately conservative and that our model captures the legitimate risk factors underlying those decisions. These results strengthen confidence that the scorecard can be reliably extended to new applicant populations with limited credit history.

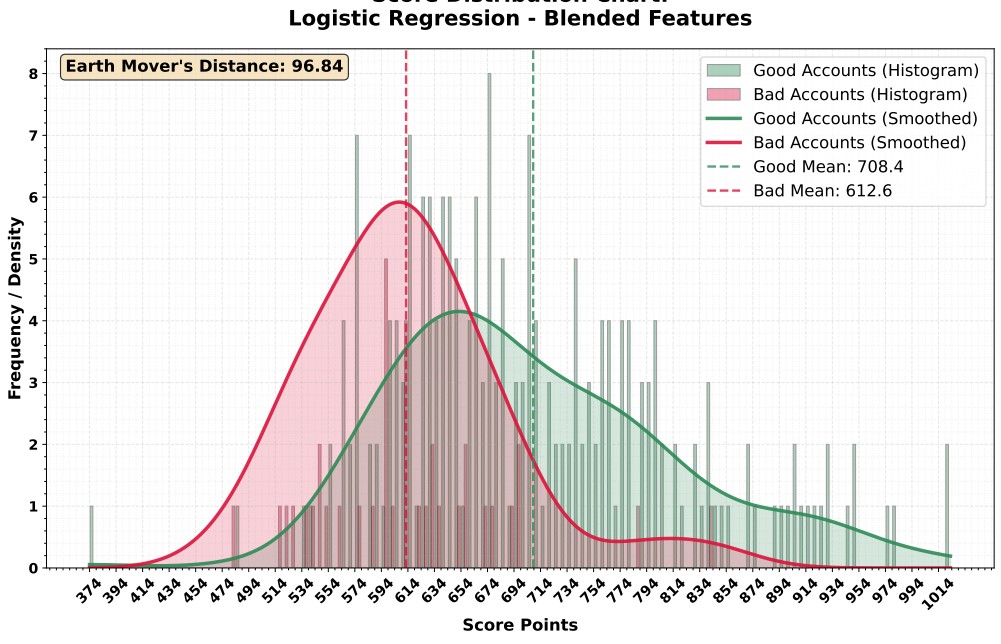

Figure 5: Score distribution for the bank statement model. Bank statement-derived cash flow indicators produce stronger separation with good accounts (mean 699.2) and bad accounts (mean 625.7) showing clearer divergence. EMD of 74.48 (3.1× higher than application information) reflects the discriminatory advantage of transactional data.

Figure 6: Score distribution for the blended model. Combining application and bank statement features achieves the strongest separation with good accounts (mean 708.4) and bad accounts (mean 612.6) showing maximal divergence. EMD of 96.84 (4.0× higher than application information alone) demonstrates complementary value of both feature sources.

# B  Deployment and Operational Framework

The sixth phase of CRISP–DM focuses on the deployment of the proposed cash flow underwriting workflow in a production environment. This workflow is designed to leverage verifiable transaction data to enable transparent and evidence-based credit decisions while establishing a continuous feedback cycle for data collection, model retraining, and performance monitoring within existing core banking systems. To support this, a robust machine learning operations framework is implemented to structure the end-to-end pipeline into five key stages. Lastly, an integrated credit scoring framework is introduced to provide comprehensive risk evaluation for both established and new-to-credit MSMEs.

## B.1  Data Ingestion

The pipeline begins when bank statements are received. Extraction modules parse and structure the raw transaction data using OCR and layout analysis techniques. This process transforms unstructured documents into a standardized tabular format suitable for downstream processing. The ingestion stage also performs integrity verification and data validation checks to ensure document authenticity.

## B.2  Feature Engineering and Feature Store

Following ingestion, structured data are fed into the feature engineering module, which computes transaction-derived and behavioral features. These features are time-stamped, versioned, and stored in a centralized repository. The feature store provides a single source of truth for both training and inference, ensuring the same logic used to generate historical features is consistently applied to new applicants in real time. This design guarantees reproducibility, prevents feature skew, and supports traceability across model iterations.

## B.3  Continuous Integration (CI)

The CI pipeline automates model retraining and validation prior to integration into the production registry. It is triggered by two primary events: (1) a predefined schedule (e.g., annually) when new labeled data become available from the core banking system, and (2) alerts from the monitoring system indicating potential model or data drift. Upon activation, the pipeline retrieves the latest versioned feature set and ground truth labels, retrains the model, and evaluates its performance against the current production model using predefined and consistent evaluation metrics.

## B.4  Model Registry and Continuous Deployment (CD)

If the retrained model demonstrates superior performance, it is automatically versioned and stored in the model registry. The CD process then packages the validated model into a containerized microservice and deploys it as a secure REST API endpoint. To ensure reliability and minimize production risk, a canary deployment strategy is applied—initially routing a small fraction of live traffic to the new model for performance validation before full-scale rollout.

## B.5  Continuous Monitoring and Feedback Loop

Once the bank statement-based credit scoring model is deployed, it operates within a Champion–Challenger framework (Kim et al., 2019) to enable continuous performance optimization. The Champion model serves as the current production baseline, while one or more Challenger models are periodically retrained on the latest data and evaluated in parallel. Model performance is continuously monitored and compared across predefined metrics. When a Challenger consistently outperforms the Champion, the system automatically notifies designated bank officers to review the results and promote the new model to production. This approach ensures sustained model improvement, robustness, and operational stability.

## B.6  Integrated Credit Scoring Framework

In production, the cash flow underwriting system functions as an integrated decision engine embedded within the lending process. Upon submission of bank statements, AI modules automatically extract and structure the raw data using OCR and natural language processing models. The processed data then undergoes fraud detection, cash flow analysis, and network analytics to generate transaction-derived features and credit scores for MSME applicants. The existing bureau-based scorecard, designed for borrowers with established credit histories, operates in parallel with the newly developed cash flow–based model. Each model independently produces a risk rating, and a risk override mechanism ensures conservative decisioning; if either model indicates higher risk, the final classification adopts that rating. This framework provides a transparent, evidence-based,

and data-driven approach to MSME credit assessment using adaptive, explainable AI scoring. It also extends financial access to new-to-credit businesses through verifiable transaction data.

## C  LIMITATIONS AND FUTURE WORK

This work has several limitations that should be addressed in future research. First, the dataset comprises 611 loan applications from a single Malaysian lending institution. While the dataset represents real-world production data collected over a two-year period, the sample size is constrained by the practical realities of MSME lending in emerging markets, where data availability remains limited due to stringent data privacy and regulatory requirements. Future work should validate these findings across multiple institutions and larger datasets as additional MSME lending data become available.

Second, the class imbalance in our dataset (518 non-default vs. 93 default cases) reflects the natural distribution observed in real-world lending portfolios, where the majority of borrowers successfully repay their loans. While this imbalance poses modeling challenges, it accurately reflects real-world credit risk settings. In this work, we addressed this limitation through appropriate model selection, WOE-based feature engineering, and careful evaluation using AUROC, which is robust to class imbalance. It is important to note that class imbalance is not a limitation that can be fully eliminated, but rather an inherent characteristic of credit scoring datasets. As credit assessment models improve and lending decisions become more accurate, successful repayment rates increase, thereby maintaining or even intensifying this natural imbalance. Nevertheless, future studies could explore advanced resampling techniques or cost-sensitive learning methods to further enhance model performance on minority class prediction.

Third, while we describe multiple AI modules for document processing, fraud detection, and transaction analysis within the proposed end-to-end cash flow underwriting workflow, the evaluation focuses on overall credit scoring performance rather than the effectiveness of individual modules. However, detailed module-level assessment is constrained by the use of proprietary methods and sensitive operational data that cannot be publicly disclosed. Future work utilizing synthetic or anonymized data could enable more granular module-level analysis and benchmarking. Additionally, future research could explore formalizing these modules as fully autonomous agents, enabling the development of a multi-agent credit scoring architecture with clearer inter-agent responsibilities and interactions.

Fourth, the proposed workflow has been deployed in a production environment with one Malaysian lending institution. While this demonstrates real-world applicability, broader validation across different lending institutions, regulatory environments, and MSME segments would strengthen the generalizability of our findings. Further research is needed to systematically integrate the proposed cash flow underwriting workflow with existing credit assessment systems that rely on behavioral and credit bureau data, ensuring coherent and robust risk evaluation. Additionally, longitudinal studies examining model performance stability across economic cycles would provide valuable insights into the temporal robustness and adaptability of bank statement-based features under varying macroeconomic conditions.

