# OpenReview forum: "AI-BAAM: AI-Driven Bank Statement Analytics as Alternative Data for Malaysian MSME Credit Scoring"
_ICLR.cc/2026/Workshop/AFAA — Submitted to AFAA 2026_

### Official Review · Reviewer_rRfh · 2026-02-21
**Promising application paper with fairness motivation but missing fairness analysis**

**Rating:** 3
**Confidence:** 4

**Summary:**

The paper proposes an end-to-end pipeline for MSME credit scoring in Malaysia using bank statement data as an alternative to traditional credit bureau information. They introduce a new dataset of 611 loan applicants, benchmark OCR extraction methods, and show that bank transaction features substantially improve default prediction (AUROC 0.806 vs. 0.647 with application info only).

**Strengths:**

The core motivation is a good fit for this workshop — credit exclusion of thin-file MSMEs is a real equity problem, and using behavioral transaction data to expand access is a legitimate fairness intervention. The dataset release is probably the most valuable contribution here, since data scarcity is what's blocked this kind of research in Southeast Asian markets. The reject inference analysis is a nice touch and shows the authors are thinking about the excluded population, not just the approved one. The OCR benchmarking is thorough and practically useful, even if it's a bit tangential to the fairness angle.

**Weaknesses:**

The paper is motivated by fairness but doesn't actually do any fairness analysis. There's no breakdown of model performance by industry, firm size, geography, or any other subgroup. For a workshop on algorithmic fairness, that's a real gap — you can't claim a system promotes financial inclusion without checking whether it works equitably across the groups you're trying to include.

The dataset is also quite small (93 defaults, 37 in validation), and no confidence intervals are reported anywhere. Some of the AUROC differences between models probably aren't statistically significant at this sample size.

Questions
Did you look at performance differences across industry sectors or geographic regions? Even a basic subgroup breakdown would go a long way here. Also, how does the system handle disagreement between its AI modules, and what are the fairness implications of the conservative risk override?

The paper tackles a real problem, contributes a useful dataset, and describes a working deployed system — all of which are valuable for this workshop's applied track. But I'd encourage the authors to add even a basic fairness analysis before the camera-ready version. Right now the fairness motivation and the technical content don't quite connect.

---

### Official Review · Reviewer_xTjU · 2026-02-21
**AI-Driven Statement-Transaction Analytics for MSME Credit Scoring in Malaysia: End-to-End Cashflow Underwriting and Cross-Financial-Institution Extraction Benchmarking.**

**Rating:** 4
**Confidence:** 5

**Summary:**

This paper studies MSMEs credit scoring in Malaysia using bank statement transactions as alternative data and proposes an end-to-end cashflow underwriting workflow that spans (i) document/key-field and transaction-table extraction, (ii) transaction feature engineering, and (iii) predictive credit scoring. The modeling results show that transaction-derived features capture dynamic financial behavior not present in application data: transaction-only models outperform application-only baselines, and the combined feature set performs best (reported AUROC up to 0.806 under in-time validation). Beyond scoring, the paper presents a broad extraction benchmark covering 30+ OCR and LLM/VLM configurations across bank statements from six banks. The authors report that a template-matching approach achieves perfect accuracy on key information extraction and the best table extraction quality (highest NED/F1) while processing documents in <0.12s with zero API cost, whereas LLM pipelines are slower (0.81–11.92s) and incur non-trivial batch cost (up to $0.53), and end-to-end vision-language models show very high latency (104–150s) with worse extraction quality. The paper positions these results as evidence that a bank-specific template strategy offers a superior accuracy–efficiency trade-off for Malaysia statements, enabling scalable underwriting.

**Strengths:**

A major strength is the full-stack framing of the underwriting system. Many papers assume clean structured transactions; here, extraction is treated as a first-order problem and evaluated comparatively. The extraction benchmark is a meaningful applied contribution: covering multiple banks and a wide set of OCR/LLM/VLM configurations produces practical evidence about which approaches are viable under latency and cost constraints. The reported finding—that tailored template matching can dominate general-purpose LLM pipelines on both accuracy and throughput for Malaysia bank statement formats—is plausible and operationally important, particularly for emerging-market lenders where unit economics and offline/air-gapped constraints matter. On the modeling side, the empirical result that transaction features add material incremental signal beyond application data aligns with how MSME risk manifests (liquidity dynamics, volatility, and concentration), and the transaction-only vs application-only vs combined comparisons provide a clear narrative. The explicit latency/cost comparisons also strengthen the feasibility claim by tying technical choices to production realities.

**Weaknesses:**

The enlarged scope (extraction + scoring) increases the need for rigorous time semantics, robustness, and governance, and those aspects remain the main areas to strengthen. First, the credit scoring evaluation is still described primarily in-time; transaction-based models are highly sensitive to temporal leakage and drift, so the paper needs explicit “as-of” feature computation and out-of-time testing to demonstrate decision-time validity. Second, results from a single lending institution can encode selection effects; label definitions, censoring/maturity, and approval bias should be clearly stated and stress-tested across cohorts and time. Third, AUROC improvements should be complemented by calibration and threshold-level decision analysis to show underwriting utility and inclusion impact at realistic operating points. Fourth, the template matching extraction results are strong but raise questions about generalization and maintenance: how brittle is template matching to bank form revisions, low-quality scans, multilingual artifacts, and edge cases? What are the monitoring and fallback strategies, and how does extraction error propagate into credit decisions? Finally, for AFAA fit, fairness/harm analysis needs to be more formal: bank transactions can encode sensitive proxies and commingling; “financial inclusion” should be defined quantitatively and tested via groupwise error/cost impacts (where feasible) and mitigation strategies.

---

### Official Review · Reviewer_NAiY · 2026-02-22
**Review: AI-BAAM: AI-Driven Bank Statement Analytics as Alternative Data for Malaysian MSME Credit Scoring**

**Rating:** 1
**Confidence:** 5

**Summary:**

This paper proposes AI-BAAM, an end-to-end cash flow underwriting pipeline that leverages bank statement transaction data as alternative data for MSME credit scoring in Malaysia. The authors introduce a dataset of 611 loan applicants from a Malaysian lending institution, develop credit scoring models using logistic regression with WOE/IV feature engineering, and benchmark 30+ OCR/LLM configurations for bank statement data extraction. The central claim is that incorporating bank statement-derived features improves credit scoring performance, with a reported AUROC of 0.806 (blended features) versus 0.647 (application information only).

**Strengths:**

- The problem of MSME financial inclusion in emerging markets is practically important
 - The extraction module benchmarking across 30+ OCR/LLM configurations and 6 Malaysian banks represents significant engineering effort.
 - The promise to release an anonymized Malaysian bank statement dataset could benefit the applied FinTech community.

**Weaknesses:**

- **Misalignment with workshop scope.** The AFAA workshop focuses on algorithmic fairness, alignment procedures, and agentic AI systems. This paper engages with none of these themes. There is no fairness analysis, no bias examination, no discussion of alignment, and no agentic system design. The brief mention of agentic workflow in the related work section is a literature citation with no connection to the proposed system, which is a conventional sequential ML pipeline. Crucially, the dataset contains features  that would readily support fairness analysis across MSME subgroups (e.g., geographic region, minimum director age), yet this is unexplored. The paper would be better suited for an applied FinTech or banking technology venue.

- **Simplistic methodological without novelty.** Logistic regression with WOE/IV binning is textbook credit scoring methodology that has been industry standard for decades. Nothing in the modeling, feature engineering, or system design appears to have any novelty. Overall, the paper reads more like a business case study rather than a research contribution.

- **Statistically unsupported central claim.** The paper's primary finding, that bank statement features improve predictive performance, rests on 93 total default cases, with only 37 defaults in the validation set. At this scale, AUROC differences are highly susceptible to noise and sampling variance. Yet the paper reports no confidence intervals, no bootstrapping or significance tests, and no standard errors. Claims such as "24.6% improvement" are presented as definitive findings without any statistical backing. This is a serious methodological shortcoming that undermines the paper's core contribution.

- **Overclaimed contributions.** The paper frames itself as introducing the "first-ever Malaysian bank statement dataset" and a novel "AI-driven" system, but the actual technical content is a standard industry credit scoring exercise with conventional tools. Overall, the paper  lacks both scientific novelty and rigor.

---

### Meta-Review · Area_Chair_DEpH · 2026-02-26

**Recommendation:** Reject
**Confidence:** 4

**Metareview:**

This paper presents an end-to-end credit-scoring pipeline that leverages bank statement data to improve financial inclusion for MSMEs in Malaysia. While the motivation for credit inclusion is noble and honest—as noted by reviewer rRfh—and the development of practical, end-to-end systems is considered important (reviewer xTjU), the submission is fundamentally out of scope for the AFAA workshop. Reviewers NAiY and rRfh correctly point out that the paper is currently a technical application study that lacks any formal fairness analysis or evaluation of bias, which are the core requirements for this venue.

The main weakness lies in the disconnect between the social motivation of "inclusion" and the technical execution, which focuses solely on predictive performance. While this work would be a strong candidate for application-focused venues, it requires a rigorous fairness assessment to be suitable for future iterations of this workshop. The authors are encouraged to incorporate fairness metrics and bias detection into their pipeline, as such an analysis would make this a compelling and impactful contribution to the algorithmic fairness community.

---

### Decision · Program_Chairs · 2026-03-02

Reject